*Revised manuscript submit to NHESS*

# Flood Vulnerability and Risk Assessment of Urban Traditional Buildings in a Heritage District of Kuala Lumpur, Malaysia

Dina D'Ayala[1*], Kai Wang[1], Yuan Yan[1], Helen Smith[2], Ashleigh Massam[2], Valeriya Filipova[2], Joy Jacqueline Pereira[3]

1. *Epicentre Research Group, Department of Civil, Environmental and Geomatic Engineering, University College London, UK*
2. *JBA Risk Management, Skipton, UK*
3. *Southeast Asia Disaster Prevention Research Initiative (SEADPRI), Institute for Environment & Development (LESTARI), Universiti Kebangsaan Malaysia, Malaysia*

Corresponding author: Dina D'Ayala, d.dayala@ucl.ac.uk

**Abstract:** Flood hazard is increasing in frequency and magnitude in Southeast Asia major metropolitan areas due to fast urban development and changes in climate, threatening people's properties and life. Typically, flood management actions are mostly focused on large scale defenses, such as river embankments or discharge channels or tunnels. However, these are difficult to implement in town centres without affecting the value of their heritage districts, and might not provide sufficient mitigation. Therefore, urban heritage buildings may become vulnerable to flood events, even when they were originally designed and built with intrinsic resilient measures, based on the local knowledge of the natural environment and its threats at the time. Their aesthetic, cultural and economic values, means that they can represent a proportionally high contribution to losses in any event. Hence it is worth to investigate more localised, tailored, mitigation measures. Vulnerability assessment studies are essential to inform the feasibility and development of such strategies. In this study we propose a multi-level methodology to assess the flood vulnerability and risk of residential buildings in an area of Kuala Lumpur, Malaysia, characterised by traditional timber housing. The multi-scale flood vulnerability model is based on a wide range of parameters, covering building specific parameters, neighbourhood conditions and catchment area conditions. The obtained vulnerability index shows ability to reflect different exposure by different building types and their relative locations. The vulnerability model is combined with high resolution fluvial and pluvial flood maps providing scenario events with 0.1% Annual Exceedance Probability (AEP). A damage function of generic applicability is developed to compute the economic losses at individual building and sample level. The study provides evidence that results obtained for a small district can be scaled up at city level, to inform both generic and specific protection strategies.

## 1. Introduction

The Sendai Framework 2015- 2030 identifies clearly both climate change and rapid urbanisation as disaster risk drivers (UNISDR, 2015). Temperature rise and global warming are strictly correlated to increased rainfall (Min et al., 2011, Wang et al., 2017) and in turn with the increased frequency and extent of droughts and floods (Pall et al., 2011; IPCC, 2013, 2014; Mysiak et al., 2016). Flood risk however is compounded not only by intensified hazard, but very importantly by increased exposure due to increased urbanisation along coastlines, river basins and flood plains (Neumann et al., 2015, Kundzewicz et al., 2013). Such flood risk becomes even more challenging in South and Southeast Asia, as observed (Najibi and Devineni, 2018) and projected (Harabayashi et al., 2013) flood frequency show dramatic increasing trends.

Following studies on the increased flood risk caused by the increasing rate of impervious surface to drainage capacity in urban areas, (e.g. Ashley et al., 2005, Jacobson, 2011, Jha et al., 2012, Liao, 2012), the shift from control to adaptation in urban flood resilience is increasingly advocated by governmental agencies, experts and developers alike. Structural mitigation measures have the objective of reducing the hazard, i.e. the runoff, by diverting it and channelling it. However, structural measures are mostly planned at large scale, require substantial investments, long implementation periods, extensive socio-political negotiation. As a consequence of this long timeframe, they might turn out to be inadequate, postponed or irreversible (Aerts et al., 2014), and in many cases they prove to be unsuitable for developing countries on economic and financial grounds (Inaoka et al., 2019). Non-structural measures, such as measures at the building scale or small-scale urban rehabilitation measures, however, can provide faster flood risk mitigation, yielding improved adaptability, (Andjelkovic, 2001; Kang et al., 2009), more distributed benefits and, as a result, better governance (Tullos, 2018). Such measures are now widely advocated by governmental and non-governmental agencies in many countries, as specifically suitable to heritage centres (Howard et al., 2017). Other non-structural measures, such as financial incentive and insurance are not investigated in this study, as there is insufficient evidence of their implementation in the study area (Roslan et al., 2019).

Studies specific to Malaysia have shown that rapidly increasing flood events in recent decades are due to unrestrained occupation of rivers by human activities, destruction of forest and extreme weather events caused by climate change (Aliagha et al., 2013). Statistics show an average of 143 floods per year since 2001, of which more than 90% are flash floods (Anip and Osman 2017). Such frequently occurring floods cause a high level of threat to Malaysian citizens' personal safety and property, thereby, inflicting considerable damage to the country's infrastructure (Nasiri & Shahmohammadi-Kalalagh, 2013). Data from the United Nations Office for Disaster Risk Reduction (UNDRR)'s Country Disaster and Risk Profile (Preventionweb 2019) show for Malaysia that floods account for 98% of average annual loss in the period 1990 to 2014. A report from the Malaysian Department of Irrigation and

Drainage (2003), identified an average of 29,000 sq.km or 9% of the country's total land area and more than 4.82 million people (22% of the population) as affected by flooding every year. The annual losses were evaluated at RM915 million (DID, 2003, accessed online 2019). At the beginning of the millennium an integrated flood management strategy was launched, whereby the Malaysian government invested in some major structural measures, along with non-structural measures and community participation. (DID, 2003, accessed online 2019). In terms of urban flood mitigation, among the structural measures, the most conspicuous intervention is certainly the SMART (Stormwater Management and Road Tunnel) project, aimed at alleviating the flooding problem in the city centre of Kuala Lumpur caused by the Klang River, as well as reducing traffic congestion (Abdullah, 2004). The SMART project is a flood diversion measure, realised as a tunnel bypass, diverting catchment discharge from the Klang Basin. Among the non-structural measures the government has also invested in flood detection and warning systems, awareness campaigns and flood proofing guidelines for buildings with basement (DID 2006; 2008). The effect of the SMART tunnel on the flood risk of the studied area is analysed in this study (See sections 2.2 and 3.3).

Notwithstanding this proactive approach, the "Malaysia Disaster Management Reference Handbook 2019" states that: "Annually, floods account for the most frequent and significant damage, with 38 damaging events in the last 20 years, and are responsible for a significant number of humans lives lost, disease epidemics, property and crop damage, and other losses". The Handbook also points out that risk of floods has increased due to climate change, stating that "Malaysia had the highest percentage of the population (67%) exposed to floods among ASEAN (Association of Southeast Asian Nations) member states between July 2012 and January 2019" (see CFE-DMHA, 2019, p 22). With six major events in the last five years, flooding remains a major source of risk and losses in Malaysia, with a dramatic three-fold increase of population exposure in two decades. While the Malaysian government has officially adopted a holistic approach to flood risk reduction from preparedness to post event relief, its implementation has received critical reviews by several researchers (Shafiai and Khalid, 2016).

Flood vulnerability, refers to the susceptibility of goods and people in any region to suffer damage and losses. An accurate assessment of such vulnerabilities is essential to devise effective flood risk management (Rehman et al., 2019). Vulnerability assessment studies, focusing on different scales (Kundzewics et al., 2019) and different dimensions (Rehman et al., 2019), have demonstrated the capability of predicting socio-economic damage and risk by floods. In an urban context, flood vulnerability assessment of individual buildings, and the management of the associated risk, has also proven to be an effective way to increase the flood resilience of the whole city (Stephenson & D'Ayala, 2014; Aerts et al., 2014). Two approaches are common in flood vulnerability assessment, the physical approach and empirical approach (Balica et al., 2013). Physical approaches use hydrological models to estimate the flood hazard and compute economic consequences for a particular event or area on the basis of a damage index relating a measure of intensity of the flood to the associated economic loss.

Parametric or empirical approaches use a set of quantitative or qualitative indicators to rate the vulnerability of a building or area, with no particular reference to the hazard intensity.

The present study is part of the 'Disaster Resilient Cities: Forecasting Local Level Climate Extremes and Physical Hazards for Kuala Lumpur', an interdisciplinary 3 years project developed through a partnership of UK and Malaysian academia, industry and local government institutions, supported by

UKRI, Innovate UK and the Malaysian Industry-Government Group for High Technology (MIGHT). The flood risk to traditional heritage houses in Kuala Lumpur, identified as one of the major contributors to disaster losses in Malaysia (Bhuiyan et al., 2018), is studied by adopting a hybrid approach using a hydrological model to determine the flood hazard and a set of indicators to determine the vulnerability of individual buildings. However, the present model does not compute the mechanical response of the

building envelop to water pressure (Custer and Nishijima, 2015).

Two different types of flooding are considered, pluvial flash flooding, caused by thunderstorms characterised by localised rainfall of very high intensity and short duration, and fluvial flooding, caused by monsoonal type long duration and low intensity rainfall over large area of the catchment. For both types of flood, the expected depths are computed for a reference 0.1% Annual Exceedance Probability

(AEP). To determine the actual risk the present study uses a multi-scale approach to assess the vulnerability of traditional houses in Kampung Baru (Figure 1), thus providing evidence to suggest appropriate mitigation strategies at individual building, local compound and district scale. The empirical vulnerability model used is particularly suitable for studies at the micro to meso scale levels, aiming at identifying effective non-structural mitigation measures. It relies on a number of quantifiable and

qualitative parameters which allow to identify construction typologies typical of the district, with diverse vulnerability level. The local elevation around the building footprint and its position with respect to any river courses are also recorded. By conducting on site and virtual surveys the parameters that influence vulnerability can be determined and quantified, and the economic losses due to flood hazards can be estimated, allowing to produce mappings which identify a ranking of risk at the building

and district scale, for a given hazard type. The hazard magnitude used is water depth, calculated by developing 2D hydrodynamic models to simulate the behaviour of water conveyed by overland flow and river systems in response to rainfall events of different frequencies and intensities. A damage function of generic applicability is developed to compute the economic losses at individual building and at sample level, considering both envelop and content damage and the loss of value associated with

the heritage character.

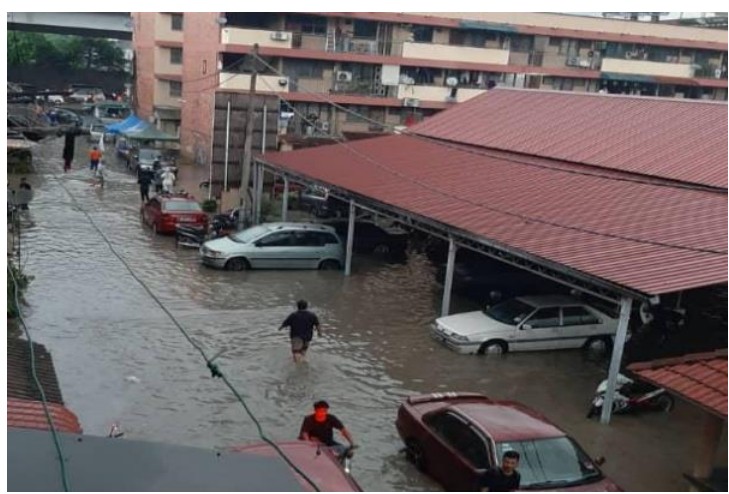

Figure 1: Pluvial Flood in Kampung Baru, 1st October 2019. Due to poor drainage, water depth of 1 meter was reached after 2 hours of rain. (BERNAMA, 2019)

## 2. Data and Methods

### 2.1 Study Area


The Kampung Baru district is located in the central area of Kuala Lumpur enclosed between the Klang River on the south east and the Sungai Bunus on the north-west (Figure 2(a)). Kampung Baru is an historic Malay Agricultural Settlement dating back more than 100 years, spread over 100 hectares and

home to approximately 19,000 residents. While having witnessed the development of the city, and being currently under pressure of gentrification, this area, which has protected status, still contains a unique building style, retaining the characteristics of both Malay traditional architecture and the ethnic Malay lifestyle. Given its setting and local topography, Kampung Baru is prone to both river flooding and flash floods, partly due to the poor drainage system (Menon, 2009; Bernama 2019) (see Figure 2).

Ju et al. (2012) recorded 121 traditional vernacular Malay houses, still inhabited by Malay people, in Kampung Baru area. These represent an important cultural and architectural heritage as well as being a touristic attraction and hence representing an important economic resource to the Malay Corporation. Although these houses might have been altered in time, in terms of materials and form, they still maintain two substantial characteristics related to the local environmental conditions: steep sloping roof

and floor raised on stilts (Figure 2(b)). These two iconic architectural features protect the space within from high intensity precipitation and frequent flooding, rendering these houses intrinsically resilient to Malay climate.

Examples of building on stilts in the area of study are shown in Figure 3. Earlier constructions are characterised by buildings on short timber stilts (3a). In some cases, the space below is enclosed by

timber grids (3b). In wealthier construction, the stilts might have been made of stone (3c) and in modern construction the stilts have been transformed in ground floor open storey (3d) to accommodate

carparking, endorsed by the Department for Irrigation and Drainage Malaysia as a non-structural flood mitigation measure.

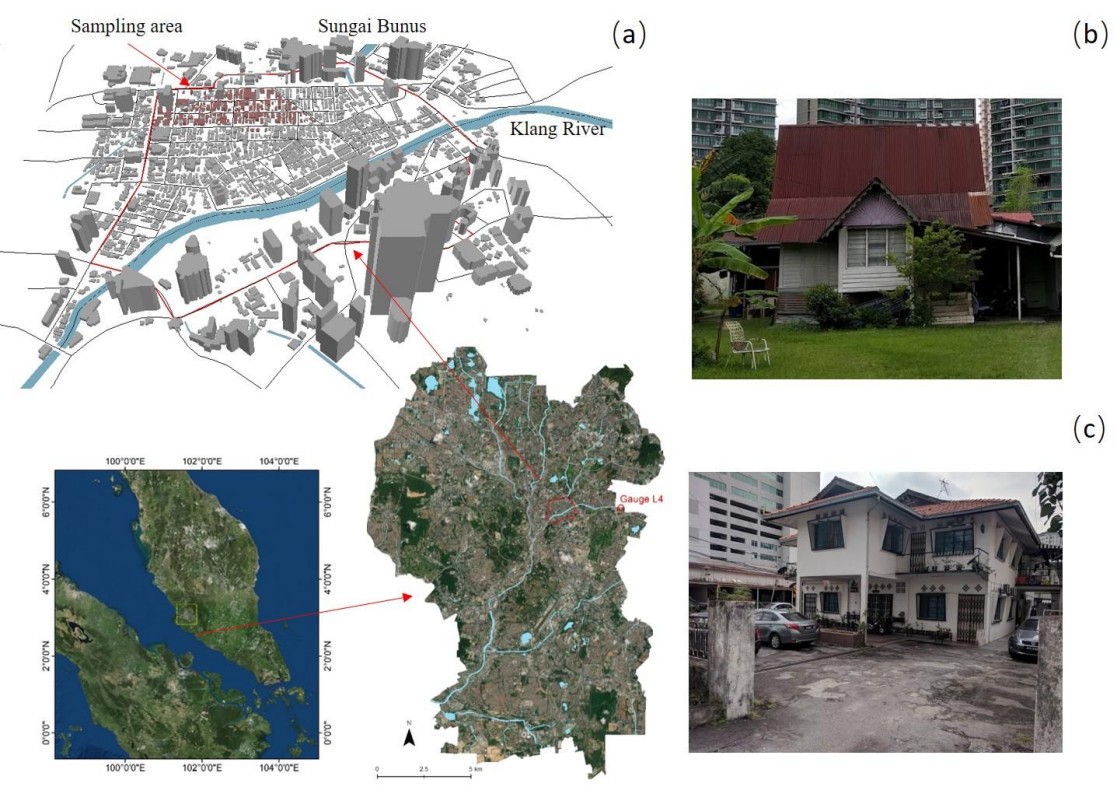

Figure 2: (a) Location of Kampung Baru in the centre of Kuala Lumpur (ESRI ArcGIS® Base Map); (b) traditional Vernacular House; (c) Modern Vernacular House.

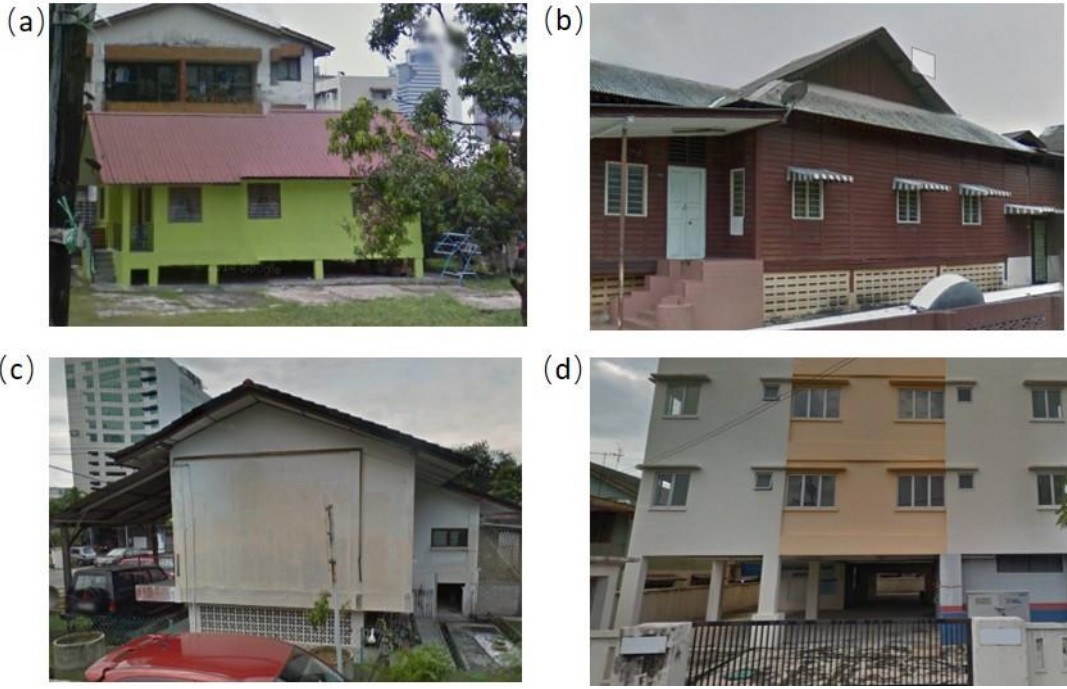

Figure 3: Typical buildings with stilts, (a) and (b) are more traditional buildings while (c) and (d) are modernized

## 2.2 Flood hazard mapping

Hazard maps showing flood extent and water depth associated with different types of flooding across Kuala Lumpur were developed within the project for a range of return periods. The maps provide water depth for pluvial flooding (also known as flash flood) and for fluvial (riverine) flooding. For fluvial flooding, two scenarios are mapped: an undefended scenario where no mitigation measures (river flood defences) are accounted for, and a scenario where the flood protection offered by SMART (see section1) is incorporated.

The maps were developed by analysing time series data from a selection of rain and river gauges across the Klang Basin to calculate intensity rainfall hyetographs and river hydrographs for return periods of 20, 50, 100 and 200-years. The intensity rainfall and river flows were used as input for 2D hydraulic modelling using JBA's proprietary JFlow® software (Lamb et al., 2009) to provide estimated depths of inundation. The methods used to calculate the rainfall hyetographs and river hydrographs are described in section 2.2.1. An important input to the flood mapping process is a digital terrain model (DTM). For this study, a 0.5m resolution bare-earth DTM was provided by the Civil Engineering and Urban Transportation Department, KL City Hall and City Planning Department, resampled to 5m resolution. This scale is commensurable with the size of individual buildings.

JFlow® can be run in different configurations for different purposes. For large rivers, a fluvial model configuration is used to apply hydrographs to the model at regularly spaced inflow points along the drainage network. The volume of water that can be held within the river channel is estimated and removed from the flood simulation. A JFlow® simulation is run for each return period using a solver based upon the two-dimensional Shallow Water Equations. For the SMART scenario a discharge-limited directional culvert is constructed in the JFlow® model, to represent the diversion and storage of flood water between Kampung Berembang and the Desa Lake at Salak South, and is adjusted for each of the four SMART operational modes as explained in Table 1.

For small rivers and pluvial flooding, a direct-rainfall configuration is used. This approach applies the relevant hyetographs to each cell of the DTM. Different runoff and drainage rates are applied to reflect spatial variations in soil type and land cover. Urban drainage systems can be accounted for by removing a proportion of the total rainfall volume prior to running the JFlow® simulation. However, in this study, no such adjustments were made as there was insufficient evidence to support quantification of urban drainage capacity across the city. Water depth in metres is calculated for each flood type (pluvial, fluvial, and fluvial with SMART defence) and return period (20, 50, 100-year) and recorded in a set of GeoTIFF raster files for use in Geographical Information Systems (GIS). In this study, flood maps of three flood types for the 100-year return period are used in the estimation of flood hazard and risk, as this is a

widely used return period in communication and decision making in flood risk prevention and management.

Table 1: Parameters of four SMART operational modes

| SMART Mode | Weather condition | Flow at stream gauge L4* | Flow diversion method | Road tunnel status | JBA return period map representing this scenario |
|---|---|---|---|---|---|
| 1 | Fair | < 70m³/s | N/a | Open to traffic | RP20-RP200 undefended |
| 2 | Moderate rainfall | 70-150m³/s | Via lower drains only | Open to traffic | RP20 defended and RP50 defended |
| 3 | Major storm | >150m³/s | Via lower drains and possibly road tunnel | Closed to traffic | N/a |
| 4 | Prolonged heavy rain | >150m³/s and Mode 3 in operation for over 1 hour | Via lower drains and road tunnel | Closed to traffic | RP100 defended and RP200 defended |

*L4 gauge is situated at confluence of Upper Klang and Ampang rivers.

### 2.2.1 Calculation of rainfall hyetographs and river hydrographs

Rainfall totals (in mm) were calculated at 11 rain gauge stations within a 6km radius of the centre of Kuala Lumpur. This was done by extracting peak-over-threshold values from the hourly rainfall record at each gauge and fitting them to a Generalised Pareto Distribution, to enable return period rainfall totals to be estimated for each gauge. This was done separately for the 1-hour, 3-hour and 24-hour storm durations. Spatial interpolation was then used to convert the estimates at the gauge stations into a set of continuous rainfall surface rasters across the entire study area, providing a rainfall total (mm) for each return period and storm duration on a 110m x 110m grid. Each gridded rainfall total was converted into a hyetograph to describe the temporal distribution of the rainfall for each of the three storm durations. Normalised rainfall profiles were developed by analysing hourly rainfall data for 20 events between 1997 and 2016 and calculating a mean 3-hr storm profile and a mean 24-hour storm profile across all stations. Due to the lack of sub-hourly rainfall data, the 1-hour storm profile was assumed to be a simple triangular shape. The storm profiles are illustrated in Figure 4(a) below.

River hydrographs were calculated at 2km intervals along the river network of the study area. Each hydrograph was constructed using a linear function, defined by peak flow and time to peak estimates. More advanced methods for deriving the shape of hydrographs are available, but in all but exceptionally flat topographies peak flow can be considered the key variable in hydrograph shape, so for this study a

generalised triangular profile was considered appropriate. Firstly, peak flow was calculated at 10
        streamflow gauges within the Klang River basin, using non-stationary flood frequency analysis. These
        values were then regionalised using a linear regression equation for each return period, enabling peak
        flow to be estimated at all ungauged locations within the study area, based on their catchment area (in
        $km^2$).

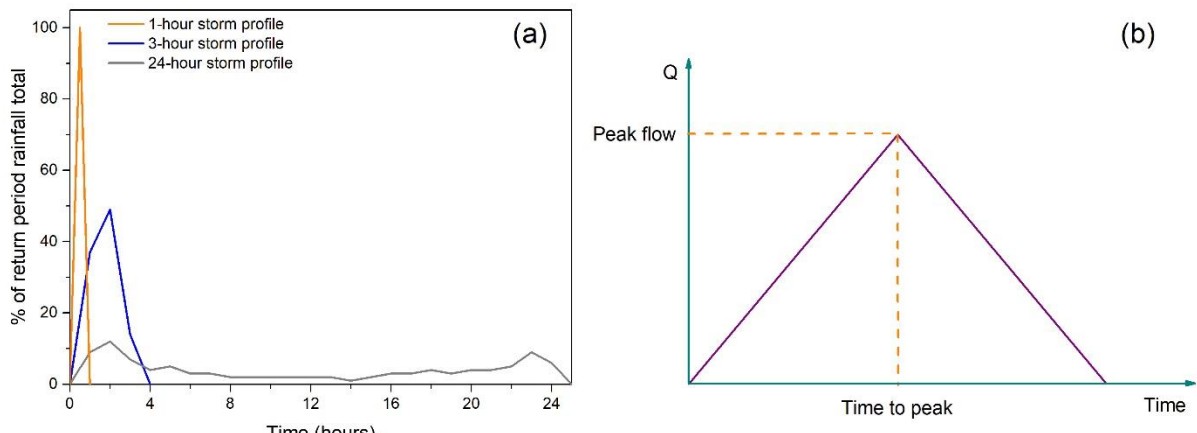


Figure 4: (a) Storm profiles used in current flood modelling (b) Schematic diagram of the river
hydrograph shape

        The time to peak at each gauge was calculated by extracting the median time to peak from all discrete
        flood events recorded at the 7 streamflow gauges with hourly flow records available. A linear regression
equation was used to estimate time to peak at all ungauged locations within the study area, which
        correlated time to peak (hours) to catchment area ($km^2$). Figure 4(b) shows a schematic diagram of the
        river hydrograph shape. Although the time to peak isn't directly relevant to the vulnerability assessment
        of buildings, it is a necessary step in constructing hydrographs which are needed to generate the hazard
        maps for different return periods.


**2.3 Data Collection**

        Given the multiscale approach adopted for the assessment of the flood risk in Kampung Baru, data is
        obtained from multiple sources. A 3D building dataset and 0.5-meter resolution DEM dataset were
        provided by UKM Southeast Asia Disaster Prevention Research Initiative (based on the 2013 LiDAR
dataset from the KL City Hall). These have been visualised in ArcMap 10.3 (Esri) and manipulated to
        extract data on building's position, footprint, position of the building's base relative to the road.  This
        information is essential to determine the depth of water at a particular building perimeter, given a flood
        depth at the site. Other data were collected from a field survey and Google Street View.  A preliminary
        overview of all buildings in the targeted area of Kampung Baru was completed on Google Street View
(GSV), to identify the most interesting sector in the district and proceed to an initial screening of the

buildings' typologies present and the identification of critical parameter to best target the field survey. The field survey of Kampung Baru, was conducted in July 2018, to gather specific data relative to individual buildings. Critical parameters, difficult to identify from the GSV, such as the location and dimensions of the drainage system, were typologically classified and measured on site, along with other

geometric parameters. A thorough photographic survey was also conducted at this stage, taking shots for all visible and accessible elevations of sample buildings, as well as larger overview shots of the whole study area. Specific features aimed at mitigating flood damage were also observed and recorded during the field survey.

After detailed data was taken on a small sample of buildings during the field survey which also allowed

for identification of buildings' typologies, a further survey based on Google Street View (GSV) was undertaken to gather additional data and cover a sample of buildings in excess of 160. This procedure was successfully used by one of the authors to survey buildings to determine vulnerability and damage in post-earthquake reconnaissance (Stone et al., 2017; Stone et al., 2018), and it is increasingly used to produce exposure databases in an expedient and economic manner (Pittore et al., 2018). In GSV, a

continuous series of 360-degree panoramas, created by sewing multiple overlapping photos together to display the real portrayal of a specific location (Street View, 2018), were observed according to the location and the time when the photos were captured. In Kampung Baru images were collated in three different years of survey, 2013, 2015 and 2017. In this study the latest version was chosen, and a full front sight of a target building could be accessed online through the observation points located on each

street. During this survey, the qualitative parameters were collected visually, replicating the field survey procedure. For quantification of other parameters, such as height of door threshold and window sills, measured samples from the field survey were used as a reference to apply a measure of scale.

**2.4 Vulnerability Model**

Research on flood vulnerability and risk assessment encompasses a wide range of methods and focuses

(Rehman et al., 2019). In an urban context a substantial component of losses is ascribable to physical damage to vulnerable buildings and their contents (Chen et al., 2016). Current flood risk assessment studies and damage models use either an empirical approach, relying on post event damage data collection to determine vulnerability functions, or synthetic approaches, whereby the vulnerability functions are based on expert opinion. Empirical methods are basin or catchment specific (Merz et al.,

2010), hence of limited transferability and applicability to other locations without substantial calibration. Synthetic models are more adaptable spatially and temporally; however, they are often based on a single variable relating flood depth to economic loss, possibly mediated by building type (e.g. HAZUS-MH, FEMA 2013). Dottori et al. (2016) present one of the few synthetic flood damage models based on a component-by-component analysis of direct damage, correlating each damage component to different

flood actions and specific building characteristics. The damage functions are designed using an expert-

based approach validated on loss adjustment studies, and damage surveys carried out for past flood events.

Historic data on flood damage and insured losses is not available for Kuala Lumpur or Kampung Baru. It is increasingly recognised that models need to account for multiscale, from single asset to full catchment area, and be able to consider many variables, in terms of both hazard intensity and asset response (Amadio, 2019). Such models may rely on sophisticated physical modelling of the flood event, while hazard-damage correlations are then determined using artificial neural networks or random forests analysis of past damage data (e.g. Merz et al., 2013; Carisi et al., 2018), or Bayesian networks (Vogel et al., 2013). For the majority of these models, however, while hazard and exposure are treated to a high level of resolution, the individual building's vulnerability descriptors are limited in number and often of a qualitative nature. Papatoma et al. (2019) suggest a method for the vulnerability indicators selection, which relies on data from systematically documented torrential events to select and weigh critical indicators using an algorithm based on random forest. Although Kelman and Spence (2003), Custer and Nishijima (2015), Hebert et al. (2018), and Milanesi et al. (2018) have used mechanical approaches to determine the structural capacity of individual masonry walls to water pressure and derive vulnerability functions which correlate physical damage to depth of water, such physical models have not so far found direct application at urban scale.

In the present study, the PARNASSUS V.3 procedure, based on a vulnerability index approach, is applied to determine the relative vulnerability of individual buildings. The building and its immediate curtilage are here defined as the system exposed to the flood hazard. Therefore, the vulnerability index is obtained by identifying a number of parameters which are considered all equally critical to the response of the system, ranging from its characteristics to its surrounding conditions. The parameters used in the present study for characterising the building vulnerability are adapted from studies conducted by one of the authors on historic buildings in UK (Stephenson and D'Ayala, 2014) and the Philippines (D'Ayala et al., 2016). Parameters such as number of storeys and footprint, provide indications on the volume of the building, its content and the bearing pressure on the ground. This has implication on soil failure and subsidence following floods, which could write off the building, hence outweighing the lower proportion of exposure of the total volume of the building, usually assumed for multi-storey buildings. This is particularly relevant for the long-term flooding scenarios. Other descriptors such as height of the base, the stilts, the door threshold and windows' sill, allow to estimate vulnerability to water breach in relation to flood depth. Finally, building's fabrics and building's condition, provide a measure of the permeability of the building construction materials and their likelihood to deteriorate when exposed to water. Besides these building-specific parameters a classification of drainage systems in the immediate setting of the buildings, of the surface condition surrounding the building and of any local flood prevention measure, are also included as vulnerability indicators. This is because typically flood hazard models, although take account of these parameters at

urban scale, by assuming certain land uses and generic drainage rates, they do not capture the local differences at the building scale. In this specific case study, as there is no sufficient knowledge of the drainage system at the city scale, such data becomes a critical indicator of vulnerability at the local scale, and one that can be directly surveyed on site. The full list of parameters is shown in Figure 5 and Table 2. The attributes for each parameter and the rating scheme adopted are described in the next section.

325

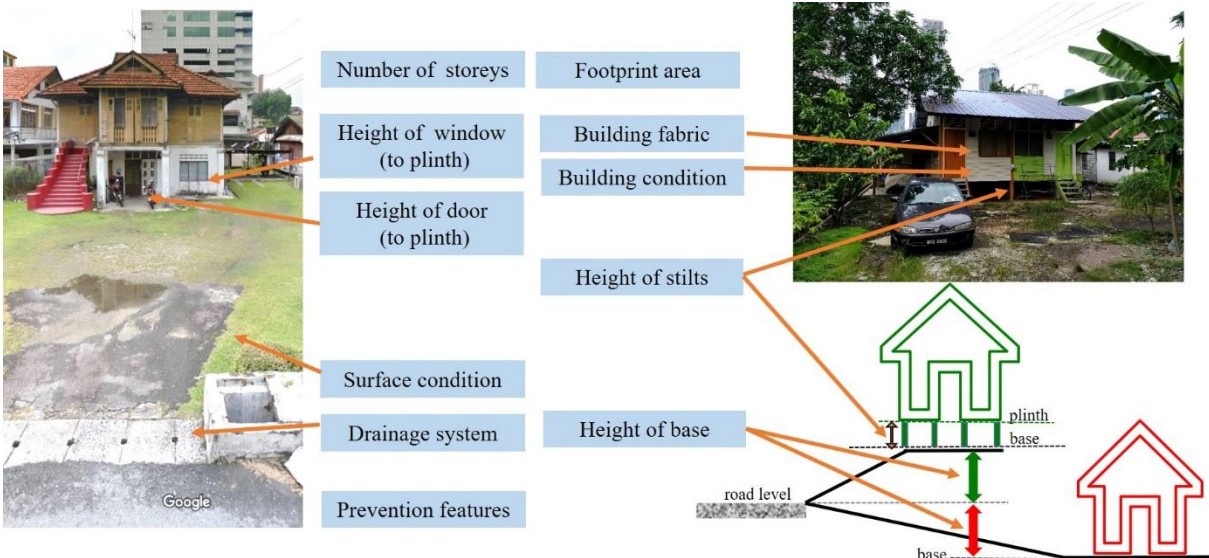

Figure 5: Example of traditional buildings in Kampong Baru and indication of the vulnerability index parameters

330

Table 2: Flood Vulnerability Index parameters for PARNASSUS V.3

| PARAMETER | DESCRIPTION | UNITS |
|---|---|---|
| 1. Number of storeys | Maximum number of storeys of the building | - |
| 2. Footprint | Building Footprint area at ground floor | m² |
| 3. Height of base | Height of the base relative to the road | m |
| 4. Height of Stilts | Stilt height over building base and position of plinth | m |
| 5. Height of door | Height of door threshold to the plinth | m |
| 6. Height of window | Height of window sill to the plinth | m |
| 7. Building fabric | Structure and cladding material | - |
| 8. Building condition | The level of maintenance and building quality | - |
| 9. Drainage system | The level of drainage system around the building | - |
| 10. Surface condition | Type of surface around the building, surface cover, inclination and permeability | - |
| 11. Prevention features | Measures of flood prevention for the building | - |

## 2.5 Vulnerability Ratings

For each parameter a range of attributes varying between 3 and 5 is determined through logical derivation of the maximum possible number of responses and these are assigned a vulnerability rating (VR) on a scale from 10 to 100. Qualitative parameters have 3 attributes and quantitative parameters have 4 or 5 attributes to ensure important measurement thresholds, affecting the building's vulnerability are captured. The scale is divided into equal, unweighted parts according to the number of attributes, with the attribute indicating lowest vulnerability assigned the value 10, and the one indicating the highest assigned the value 100, as shown in Table 2, following the PARNASSUS V.1 procedure (Stephenson and D'Ayala, 2014). For instance, the parameter 'drainage system' has three possible outcomes: 'good', 'poor' and 'no', so that the numerical rating among these three outcomes can be assigned as 10, 55 and 100, to represent the increase in vulnerability. Table 3 summarise each parameter range of attributes and its conversion into vulnerability rating. The surface condition consists of three sub-parameters and the building fabric consists of two sub-parameters. In both cases, the vulnerability rating is calculated as the average ratings of the sub-parameters.

Table 3: Parameters' attributes and corresponding vulnerability rating in PARNASSUS V.3.

| Parameter | Sub-parameter | possible outcome | VR | Parameter | Sub-parameter | Possible outcome | VR |
|---|---|---|---|---|---|---|---|
| 1. number of storeys | | >=4 | 100 | 7. Building fabric | frame material | timber | 100 |
| | | 3 | 70 | | | masonry | 55 |
| | | 2 | 40 | | | concrete | 10 |
| | | 1 | 10 | | wall material | timber | 100 |
| 2. Footprint | | >500 | 100 | | | masonry | 55 |
| | | [400, 500) | 77.5 | | | concrete | 10 |
| | | [300, 400) | 55 | 8. Building condition | | poor | 100 |
| | | [200, 300) | 32.5 | | | good | 55 |
| | | <200 | 10 | | | excellent | 10 |
| 3. Base | Height of base to road | <-1 | 100 | 9. Surface condition | vegetation | no | 100 |
| | | [-1, 0) | 77.5 | | | poor | 55 |
| | | 0 | 55 | | | good | 10 |
| | | (0, 1] | 32.5 | | inclination | concave | 100 |
| | | >1 | 10 | | | flat | 55 |
| 4. Stilt | Height of stilts | 0 | 100 | | | convex | 10 |
| | | (0, 0.5) | 55 | | permeability | no | 100 |
| | | >0.5 | 10 | | | poor | 55 |
| 5. Door threshold | door to plinth | 0 | 100 | | | good | 10 |
| | | (0, 0.1] | 70 | 10. Drainage system | | no | 100 |
| | | (0.1, 0.5] | 40 | | | poor | 55 |
| | | >0.5 | 10 | | | good | 10 |
| 6. Window sill | window to plinth | 0 | 100 | 11. Flood-prevention features | | no | 100 |
| | | (0, 0.5] | 70 | | | yes | 10 |
| | | (0.5, 1] | 40 | *12. traditional construction | | no | |
| | | >1 | 10 | | | yes | |

* factor used in equation (6)

Hence for each building and for each parameter a vulnerability rating $VR_{ij}$, can be defined, whereby $i$, ranging from 1 to 163, denotes the building ID, and $j$, ranging from 1 to 11, denotes the parameter under consideration. The vulnerability index $VI_i$ for each building is therefore computed by summation of the vulnerability rating for each parameter:

$$VI_i = \sum_j VR_{ij} \tag{1}$$

The vulnerability index for each building can range from a minimum of 110 for lowest vulnerability to a maximum of 1100 for the highest vulnerability. To compare the cumulative frequency of each parameter and its relevance to the $VI_i$, a normalised vulnerability rating of each parameter $nVR_{ij}$ and the total vulnerability index $nVI_i$ are calculated based on Eq (2) and (3).

$$nVR_{ij} = \frac{VR_{ij}}{(VR_{ij_{max}} + VR_{ij_{min}})/2} \tag{2}$$

$$nVI_i = \frac{VI_i}{(VI_{i_{max}} + VI_{i_{min}})/2} \tag{3}$$

where the normalisation is with respect to the mean value of the scoring range $\overline{VR_{ij}}$ and $\overline{VI_i}$. This normalisation also allows comparison among different samples of buildings at different sites.

To further analyse the data, buildings are grouped in four classes by dividing the vulnerability range in 4 equal parts: Very Low vulnerability $(0.1, 0.325 * VI_{max})$, Low vulnerability $(0.325 * VI_{max}, 0.55 * VI_{max})$, High $(0.55 * VI_{max}, 0.775 * VI_{max})$ and Very high $(0.775 * VI_{max}, VI_{max})$.

In this study, the $VI_i$ of the surveyed buildings are concentrated in the middle two categories. To refine the classification, the low vulnerability and high vulnerability categories are further divided into two equal parts: Low $(0.325 * VI_{max}, 0.4375 * VI_{max})$, Medium Low$(0.4375 * VI_{max}, 0.55 * VI_{max})$; Medium High $(0.55 * VI_{max}, 0.6625 * VI_{max})$ and High $(0.6625 * VI_{max}, 0.75 * VI_{max})$.

To determine the relative contribution of each parameter to the highest and lowest vulnerability index scores $rVR_j$ is calculated based on Eq(4):

$$rVR_j = \frac{\sum_k VR_{kj}/k}{\sum_i VR_{ij}/i} \tag{4}$$

where $j$ denotes the parameter considered, $k$ denotes the number of buildings in a given vulnerability class and $i$ is the total number of buildings surveyed.

## 2.6 Economic loss

The vulnerability index $VI_i$ derived in the previous section is a suitable measure to provide a scale of criticalities for particular properties in need of attention to improve their flood resilience. However,

interventions and investments, whether at the individual property-owner level or at the level of the council or district authorities, are usually justified on the basis of cost-benefit analysis. Typically, this is expressed in terms of a replacement cost function which quantify the damage in monetary value and relates it to a measure of the flood intensity, such as flood depth. (Pistrika, 2014) The computation of the economic losses caused by flood events includes different components, that can be classified as tangible costs, including the physical damage to the building and contents, interruption of work etc.., and other intangible costs, such as loss or damage to objects with sentimental or cultural value, difficult to quantify (Kreibich et al., 2014). The economic loss model proposed in this study considers the physical damage to each building and its content as it can be estimated on the basis of its specific vulnerability (see section 2.5) and a normalised damage factor $D(h_i)$ expressed as a function of the flood depth. Two different damage factors $D_b(h_i)$ and $D_c(h_i)$, for the building and contents, respectively, are used in the present study.

The physical damage to individual buildings can be calculated as the total replacement cost $E_i$

$$E_i = C(i) * D(h_i) * F_{VR}(VI_i) * A_{Ti} \tag{5}$$

where $i$ indicates the building identifier, $C, D, F_{VR}$ and $A_T$ are the construction cost per unit area of building, the damage factor, the vulnerability factor and the surface area of the building directly affected by the flood, respectively. They are derived as follows.

Building cost:

The replacement cost of buildings $C(i)$ includes two parts, the replacement cost of the building $C_B(i)$ and the replacement cost of contents $C_C(i)$.

$$C_B(i) = F_B(i) * F_H(i) * C_0(i) \tag{6}$$

where $C_0(i)$ is the estimated construction cost in the study area depending on building type and materials, $F_B(i)$ is a value factor depending on the perceived value of the building, $F_H(i)$ is a value factor depending on the historic and cultural status of the building. The value factor $F_B$ can be used to account for the depreciated cost, i.e. the current remaining value, rather than the replacement value (Huizinga et al., 2017). However, as several of the buildings in the study area are either historic or traditionally built, neither the depreciated cost or replacement cost might be appropriate to account for their cultural value. Arcadis (2019) uses a range from 2415 to 4105 RM (525 to 890 €) per square meter to compute the basic construction cost $C_0(i)$ of a detached house in Kuala Lumpur. This value includes the construction and services (electrics, hydraulics and mechanical) costs. In this study the building fabric material (timber, masonry, concrete) is used to determine the low, medium and high cost range, while the building condition (poor, good and excellent) is used to determine the values of the adjustment factor $F_B = (0.4, 0.7, 1)$, respectively. If the building is among the ones identified as of traditional construction by Ju et al. (2012), or listed as of historic value in this study survey, a factor of $F_H(i) = 1.3$ is applied to account for the additional cultural value as a touristic attraction.

Replacement cost for damage suffered by contents is also a non-negligible component of the total loss suffered by building affected by floods. Huizinga et al. (2017) and FEMA (2013) assume that the replacement cost of content typically ranges between 40 and 60% of the building cost for residential properties. However, studies at the microscale (Appelbaum, 1985; Oliveri and Santoro 2000) show that the proportion of content cost to structure cost also depends on type and quality of construction, level

of household income, etc. with a range from 15 to 60 %. Therefore, the content cost can be expressed as:

$$C_C(i) = C_B(i) * k_c \qquad (7)$$

where $k_c$ assumes values in the range (0.15 – 0.60), which is also determined according to the building condition in this study.

Finally, combining the building replacement cost $C_B(i)$ and the content replacement cost $C_C(i)$ provides the total replacement cost for each building.

$$C(i) = C_B(i) + C_C(i) \qquad (8)$$

The Flood depth-damage ratio function $D(h_i)$, is a function of the water depth $h_i$, which in this study is computed as the differential at each building site between the inundation depth $FD_i$ computed by the flood hazard model and the elevation of the building plinth above ground, i.e. the height of the stilts (or other structure raising the plinth) $HS_i$.

$$h_i = FD_i - HS_i \qquad (9)$$

Depth-damage ratio functions specific for Malaysia or Kuala Lumpur do not exist in literature, as data on losses from past events has not been systematically collected and analysed to date, notwithstanding the frequency of these, even just in the last decade (Romali et al., 2018). The derivation of synthetic depth-damage functions relies on appropriate exposure databases, ad-hoc surveys, or heuristic

information on losses. When conducting studies at micro scale, as the present one, it is important that the depth-damage ratio function used reflects the damage to single buildings, rather than aggregation at grid cell level or larger, and also reflect the actual response of each single construction to flood. A systematic review of several depth-damage ratio functions produced in literature (Appelbaum, 1985; Lekuthai & Vongvisessomjai, 2001; Dutta et al., 2003; Huizinga et al., 2017; MLIT, 2005; Pistrika et

al., 2014; Englhardt, 2019) show the relevance of parameters such as construction material and quality, number of storeys, conditions, etc, in determining the depth-damage function, leading to a non-negligible variance among the available functions. However, as the proposed vulnerability model discussed in section 2 accounts for these characteristics explicitly in the computation of the vulnerability index $VR_i$ for each building, it is appropriate to derive a mean damage ratio function, only dependent on

water depth, while the variance due to the building characteristics are accounted by the Vulnerability Factor $F_{VR}$ $(VR_i)$ in equation (5). Figure 6 shows the damage ratio function obtained as regression from the mean values of several damage functions available in literature, the associated variance for each point in the series, and the 95% confidence bounds. The regression damage function, with a coefficient of determination $R^2 = 0.846$ (significant at 0.01 level), shows very good correlation with damage

functions produced on the basis of actual damage databases, such as the ones proposed by Prettenthaler et al. not included in the regression sample (2010).

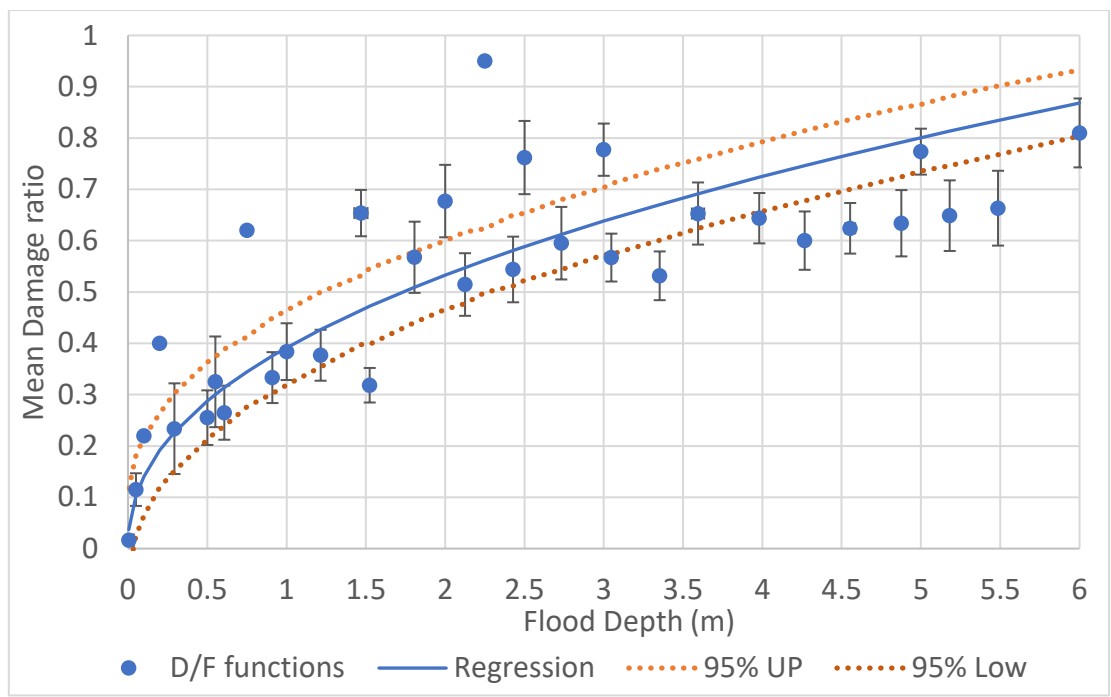

Figure 6: Mean damage ratio as function of flood depth with point by point standard deviation

 Vulnerability factor $F_{VR}$.

$$F_{VR}(VI_i) = \frac{VI_i}{VI_{median}} \qquad (10)$$

The vulnerability factor $F_{VR}(VI_i)$ for each building is computed based on the vulnerability index calculated with equation (1) divided by the median value of the distribution of vulnerability indexes in the sample of interest. In this way the replacement cost function is calibrated directly on the local

 building stock of the study area, while remaining non-dimensional and of generic validity.

Total flooded area of each building $A_t$,

$$A_{Ti} = A_{fi} * n_{fi} \qquad (11)$$

The total flooded area of each building $A_{Ti}$ equals to the foot print of the buildings $A_{i_f}$ times the number of storeys affected by the flood $n_{i_f}$, which is computed as

 $$n_{i_f} = integer\left(\frac{d_f}{h_s}\right) + 1. \qquad (12)$$

Where $d_f$ is the depth of water at the site and $h_s$ is the storey height including stilts, where appropriate

## 3. Results

### 3.1 Vulnerability Index of selected buildings

 Based on the empirical model described above, the vulnerability rating $VR_j$ for each parameter were

attributed to each building and the total $VI_i$ computed. Notwithstanding the relatively small size of the district considered, and the consequent uniformity of building height (mainly 2 storey) and footprint, Figure 7(a) and 7(b) show that the occurrence of each $VR_j$ parameter attributes and each $VR_j$ cumulative distribution, respectively, are all different, indicating that there is no direct correlation among the parameters chosen to represent the vulnerability of these buildings. Nonetheless, the $VI_i$ cumulative distribution shows good agreement with a lognormal function (Figure 7b), with a coefficient of determination 0.997 (significant at 0.01 level).

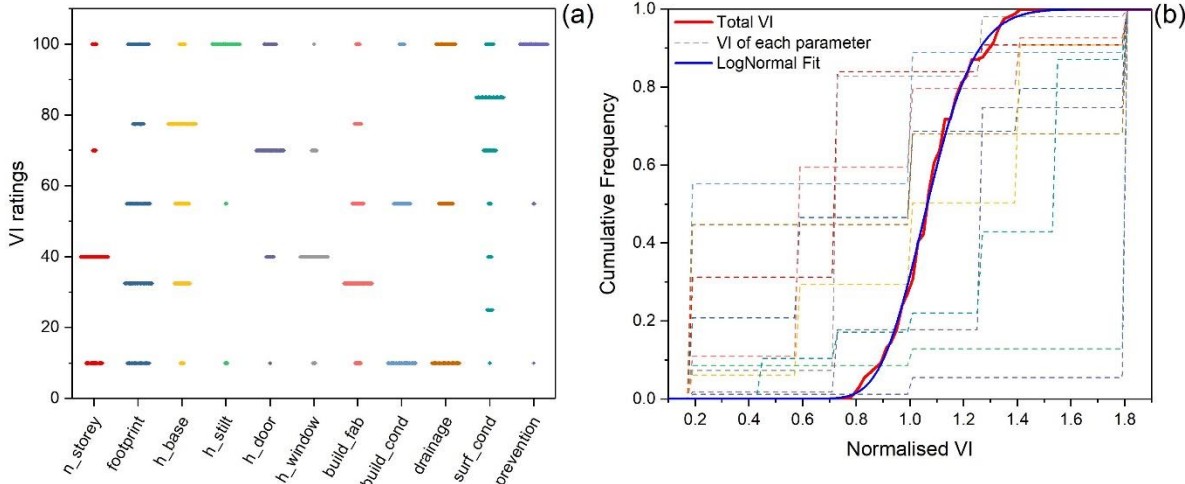

Figure 7: a) Scatter plot of the VR of each parameter b) The cumulative frequency of each parameter and the total VI for Kampung Baru buildings' sample.

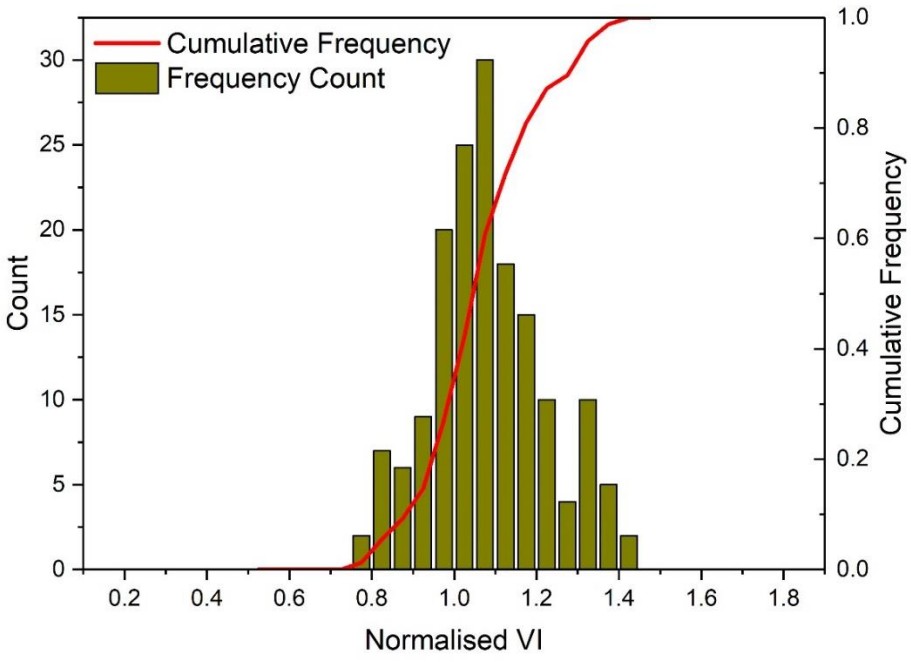

Figure 8: Distribution of normalised vulnerability index $VI_i$

Table 4 Vulnerability categories and number of buildings in each category

| Vulnerability Categories | | Quartile range VI | Percentage of value range | Occurrence in sample | Percentage in sample |
|---|---|---|---|---|---|
| Very Low | Very Low | 110-357.5 | 10%-32.5% | 0 | 0 |
| Low | Low | 357.5-481.25 | 32.5%-43.75% | 2 | 1.2 |
| | Medium Low | 481.25-605 | 43.75%-55% | 45 | 27.6 |
| High | Medium High | 605-728.75 | 55%-66.2.5% | 85 | 52.1 |
| | High | 728.75-852.5 | 66.25%-77.5% | 31 | 19.0 |
| Very High | Very High | 852.5-1100 | 77.5%-100% | 0 | 0 |

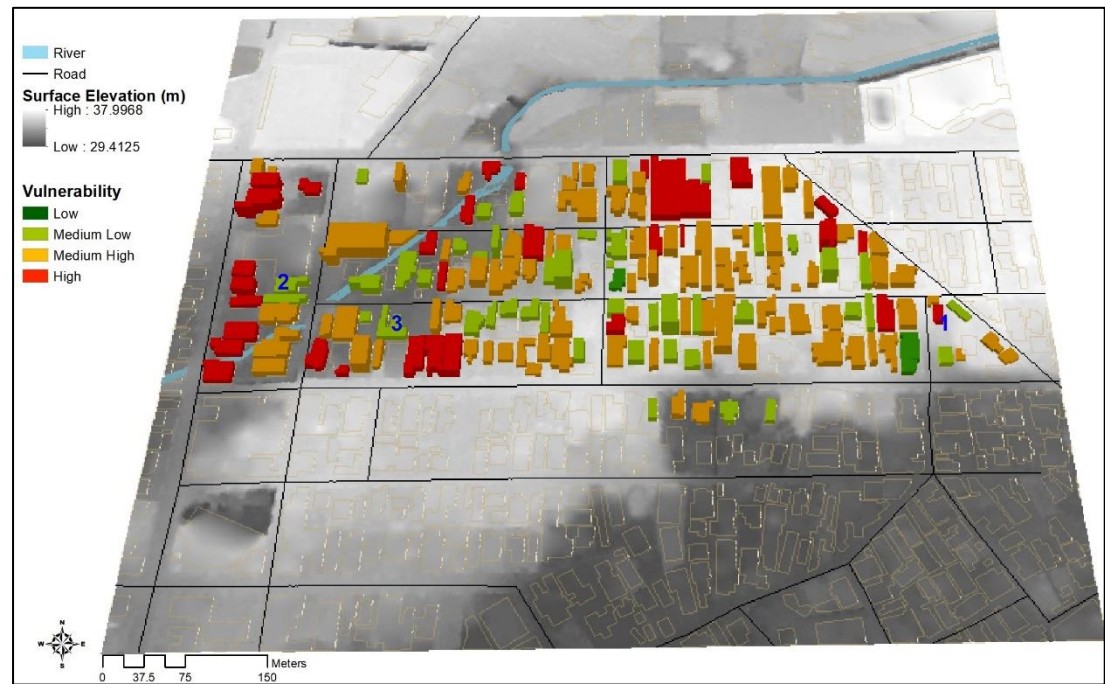

Figure 9: Spatial distribution of VR of each building. Buildings marked 1, 2and 3 are the cases described in section 3.2

The largest $VI_i$ value in the sample is 852.5, and the smallest is 477.5 (Table 4). The distribution of the values normalised with respect to the median is shown in Figure 8, together with the cumulative distribution. The full normalised range of the $VI$ is divided in four equal intervals, which determine 4 classes of vulnerability: very low, low, high, and very high, as already explained in section 2.5 and shown in Table 4. The classes low and high are further subdivided in low and medium-low, and medium-high and high, respectively. There are no buildings in the extreme classes, very low or very high vulnerability. From Figure 8 and table 4 it is evident that the overall distribution of $VI_i$ is relatively narrow, with a median greater than the average $\overline{VI_i}$ and the majority of the samples falling in the medium high vulnerability class. The low vulnerability class represent 1.2% of the sample and the high

vulnerability class includes 19% of the buildings. The spatial distribution of the vulnerability index shows a relatively random pattern, without particular alignment to the roads' grid or the relative distance from the river. (Figure 9). This confirms the lack of uniformity of the urban pattern of this district and the importance of assessing the flood vulnerability at the scale of the individual building.  As mentioned earlier, the number of storeys and footprint are relatively uniform, hence the curtilage setting and the construction details are really what characterise the variance in vulnerability. This is further explained in the next section.

## 3.2 Relevance of factors contributing to vulnerability

Given the apparent random spatial distribution of buildings in the high and low vulnerability categories, it is worth examining the relevance of the different parameters contributing to the $VI_i$ of each building, so that the adverse attributes can be mitigated to reduce risk to flood hazards. For buildings in the bottom and top quintile of the distribution, as per eq. 4, the average scoring of each parameter in that category is divided by the average scoring of the same parameter over the whole sample, hence highlighting the parameters that most contribute to the tails of the distribution. This is graphically shown in Figure 10, where 1 is the normalised value of the mean for each parameter over the whole sample. As there are only 2 buildings in low $VI$ category, another 29 buildings in the lower part of medium-low $VI$, were selected to compare with the 31 high $VI$ buildings. It is shown that for the high vulnerability class, poor drainage system, and building's condition, both have a value more than 50% greater than the average score, representing the most substantial contribution to high values of $VI_i$. The height of the base also contributes to the higher $VI_i$, in accordance with the observation that often houses are built below the road level at a distance from the drainage system and hence are located in concave, undrained settings. This condition is particularly vulnerable in the case of high intensity- short duration pluvial floods. Conversely, good drainage system, presence of stilts on the ground to elevate the plinth height, as well as good building conditions, are key parameters in low vulnerability scoring.

Further three specific buildings are selected, one located in the eastern part of the district, falling in the high class of $VI_i$; the other two located in the western region of the district, characterised by a low value of $VI_i$ (Figure 9). For the first case, the parameters that determine the high vulnerability are the lack of stilts, the poor building condition and permeable building materials, the lack of proper drainage and prevention measure, the setting of the building below the road level, although the curtilage of the building is characterised by a permeable and absorbent surface conditions. Topographically however, the building is set in the highest terrain of the district, and hence might be exposed to lesser hazard than other buildings.  On the contrary, for the two low $VI$ cases, although located in the portion of the district at lower topographical elevation and near the river, hence being characterised by high exposure, they are set at the same or higher level as the road or well above, both have door threshold set above average,

both have good drainage, and finally they either have stilts or good prevention measures, to be overall less vulnerable, or better, more resilient to the flood hazard

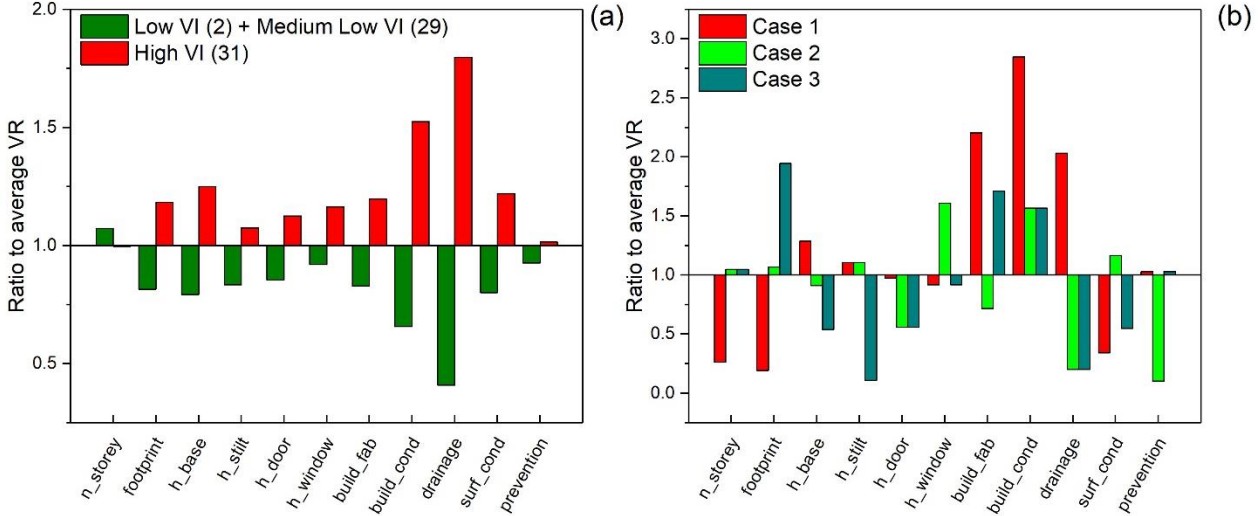

Figure 10: Relative values to the average *VI* for each parameter, (a) for the lower and upper quintile of the sample; (b) three selected cases as located in Figure 9

This is a relevant finding, as commonly, for studies at mesoscale, it is assumed that parameters such as drainage and surface conditions can be assumed as uniform over an urban block, for instance. In relation to Kampung Baru the spatial distribution of the results demonstrates that the provision for drainage and permeable ground surfaces, might be rather fragmented, even along the same street, in parts owing to plots redevelopments at different times. This further highlights the significance of local scale prevention to reduce the flood vulnerability and risk.

### 3.3 Estimation of replacement cost due to different flood scenarios

To estimate the flood damage to buildings, as introduced in section 2.2, three different scenarios are considered: a pluvial flood, a fluvial flood without structural defences and a fluvial flood considering the effect of the SMART tunnel defence (Abdullah 2004). For all scenarios the reference rainfall with 10% probability of exceedance in 100 years is considered here and the extent of flood water for each scenario is presented in Figure 11 a)-c), together with the total losses (risk map) associated to  d) fluvial flood without SMART system in operation, e) fluvial flood with SMART system in operation, f) pluvial flood.  The number of buildings flooded and economic loss as a function of water depth at each building are reported in Figure 12 where the water depth is defined as the difference between height of plinth above ground and inundation depth, which provides a direct measure of the water depth entering the buildings (Equation 9).

For fluvial flood, the flooded buildings are mostly located in the west part of the study area which is close to the Sungai Bunus river. The maximum water depth is around 1.4 m, reducing to around 1m with the action of SMART. The SMART has limited effect to flooding extent in the specific area of study, as it mainly operates on the larger Klang river. For the pluvial flood, most buildings are flooded to less than 0.2 meter, and have a scattered distribution across the study area. Notwithstanding the differences in depth and spatial distribution of the three scenarios the total number of buildings affected varies little, between 20% and 24% of the total number of buildings surveyed in the study area (Figure 12a). Note that buildings on the south-east portion of the map, close to the Klang river, are also suffering fluvial flood; however, these buildings are outside the area of the present study.

The total replacement cost is calculated based on section 2.6. This amounts to around 5M RM (≈1M €) for pluvial flood for the 163 buildings. For river floods, the total cost is considerably higher, around 15M RM (≈3M €) without defence and 10M RM (≈2M €) with SMART in operation. The percentage of cost to the total replacement cost are around 1.6%, 4.7%, and 3.1% for pluvial flood, river flood and river flood with SMART respectively. The majority of economic losses for pluvial flood are concentrated around 0.2m water depth; for fluvial flood without SMART the majority of losses are concentrated in the range between 0.5 to 1.4 m; finally, for fluvial floods with SMART, losses are distributed mainly around 0.5m to 0.7 m with a maximum of 1.1.m. Figure 12a also shows a number of building with negative water depth: these are buildings with stilts, where the flood depth is lower than the position of the plinth above ground, meaning that although the buildings curtilage gets flooded, this does not affect the building itself. This corresponds to 6% of the present sample. To emphasise the relevance of the accurate elevation of the point of first breach in the building, i.e. the vertical position of the door threshold with respect to the ground, Figure 12c shows the difference in total losses for each of the 3 scenarios considered. The reduction in total losses ranges from a minimum of 13% for the fluvial flooding with the SMART activated scenario, to a maximum of 20% for the flash flooding scenario. Figure 12c also shows the range of variability of the total losses when the 95% confidence bounds of the damage ratio function are considered.

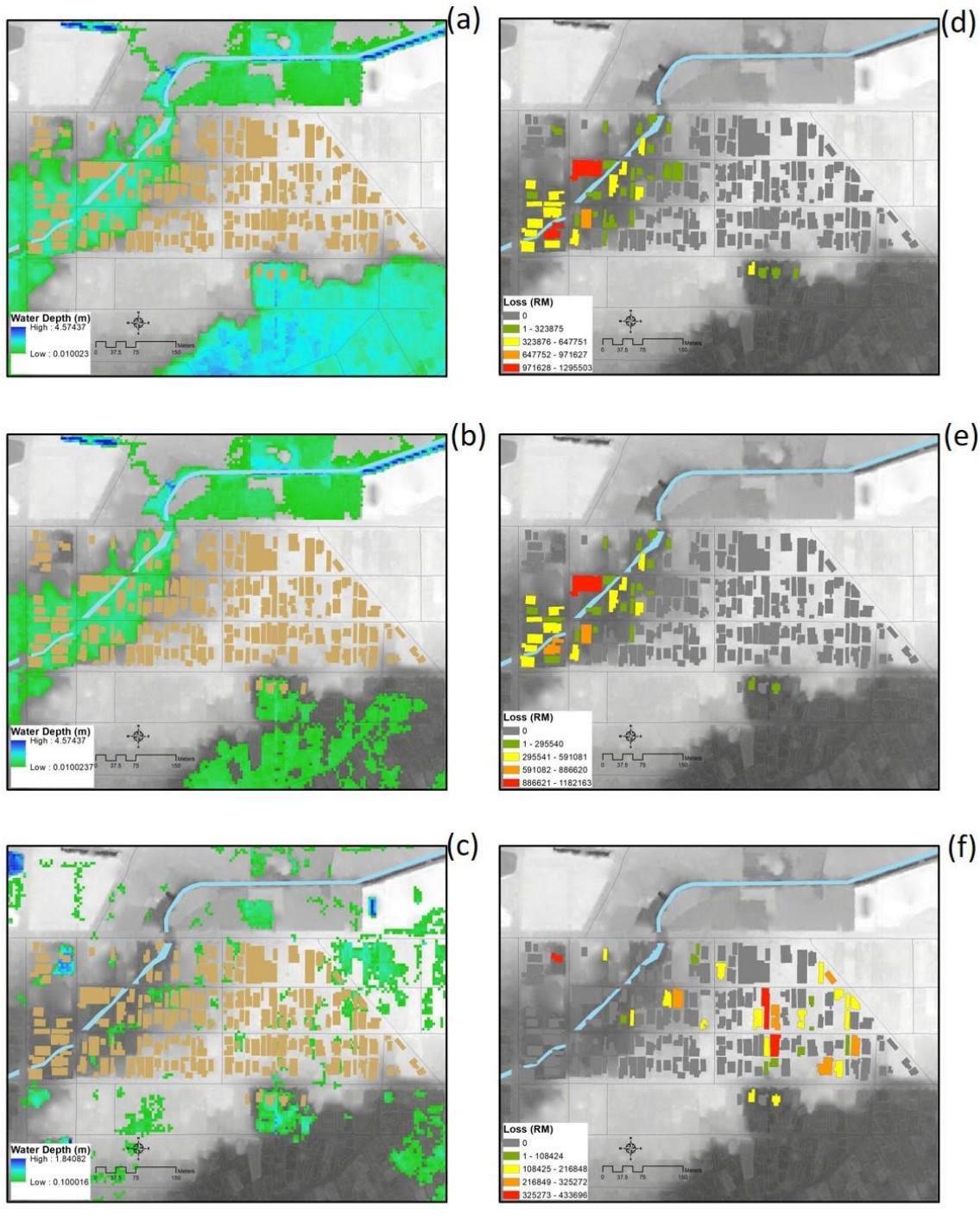

Figure 11: Flood Maps of different scenarios (a) River flood without SMART (b) River flood with SMART (c) Flash flood, and the estimated total replacement cost due to river flood without SMART (d), with SMART (e) and flash flood (f). All under100 year return period.

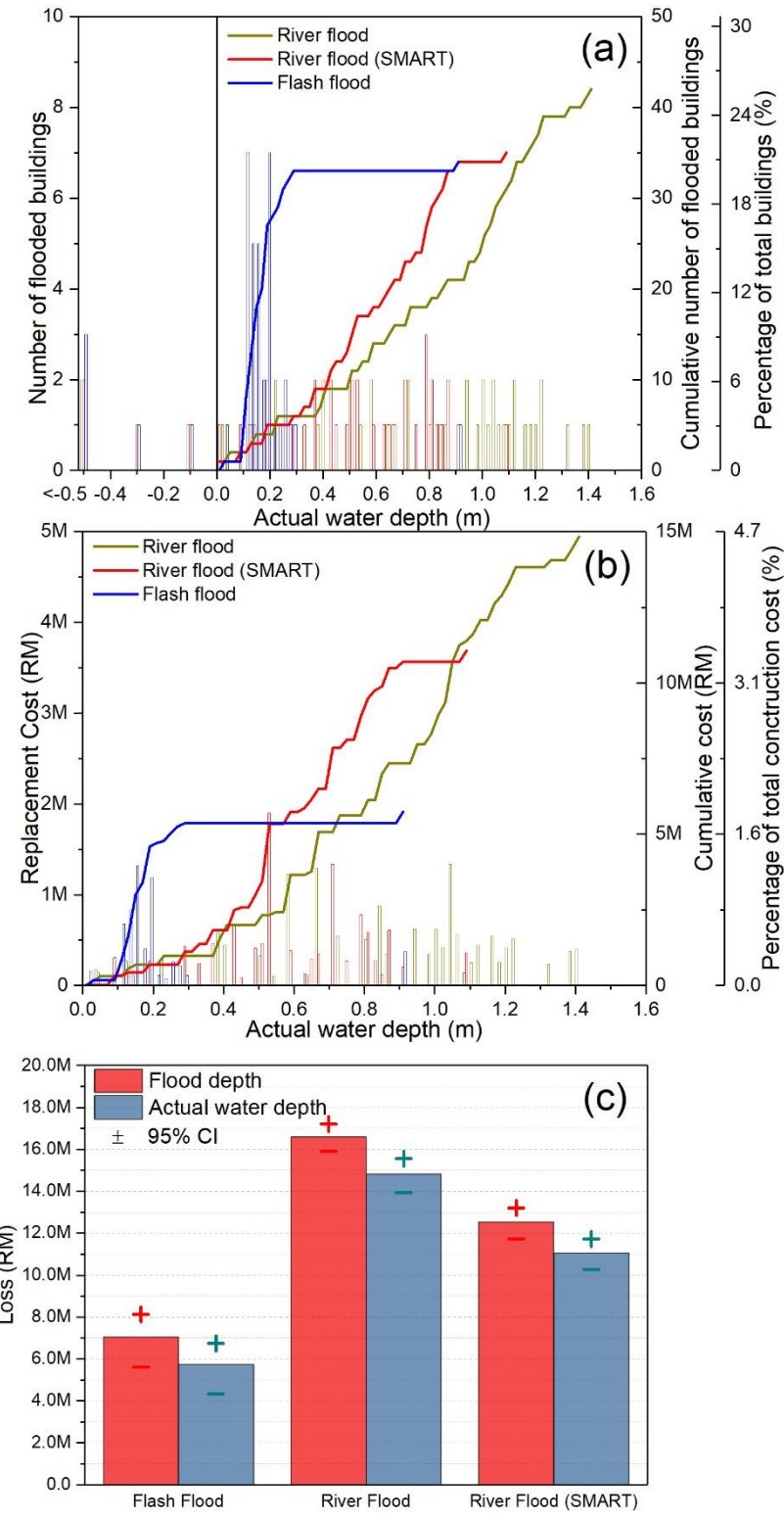

Figure 12: Number of flooded buildings (a) and total replacement cost (b) for different flood scenarios. Some buildings with stilts get flooded but have no damage, hence are reported as having negative actual water depth. (c) The calculated difference in the loss between flood depth and actual water depth.

**4. Discussion**

While major improvements in modelling flood hazard and exposure have been achieved, there is still a lack of compelling evidence on spatio-temporal patterns in vulnerability of societies around the world (Jongman et al., 2015). The Southeast Asian region is more vulnerable due to the higher population density and higher frequency of rainfall. This study focusses on flood vulnerability of the buildings in a small heritage community, Kampung Baru, in the city centre of Kuala Lumpur, Malaysia. This city has experienced an increasing number of flood events due to the combined effects of observed increasing extreme rainfall referred to as Wet Wetter Dry Drier pattern (Allan 2008, 2010) as well as an increase of urban population, nearly doubled from 1980 to the current 1.8 million . As the trends for these two variables are not slowing or reversing, it should be expected in the future that both flood hazard and exposure in this city will continue to increase.

Buildings, being the primary shelter for people, the reduction of their vulnerability is critical in reducing the risk to flood faced by population. By determining and quantifying the value of vulnerability and risk for each building exposed to specific flooding scenarios, these can be visualised on thematic maps, thus providing evidence to suggest appropriate design or protection strategies specific to each building in the area of study. The present study has identified that higher vulnerability is related to absence or poor drainage system, poor building's conditions and poor overall surrounding surface conditions. The buildings with lowest vulnerability show a combination of good drainage systems and surface condition and/or stilts at the ground floor or other forms of protection. The lognormal vulnerability cumulative function obtained has generic validity and it is a synthetic representation of the vulnerability of the district which can be used at different levels. For building owners, $VI_i$ can be used to determine the level of vulnerability of their property and identify features that can be improved to reduce such vulnerability. At the level of the district and with reference to the map as well as to the division in vulnerability classes, it can be seen that buildings belonging to the same class are clustered, meaning that there are local intervention at the scale of few compounds, (such as drainage, surfacing, slope) which can be address to reduce such vulnerability. At the municipal level, if this exercise is repeated for different neighbours and districts then a ranking of them in relation to the mean and dispersion of the $VI$ function can provide support to decision making in terms of non-structural flood defences at neighbourhood scale. Thus, several possible solutions can be provided to improve the flood vulnerability of building in Kampung Baru or similar districts, among which some feasible strategies are:

1. Increasing the ground floor base elevation by either adding pillars or stilts at ground level in new design . The raising floor on stilts is a traditional design of Malaysian vernacular buildings, common of many surveyed cases in Kampong Baru, and such design is being modernised by introduction of open car park at the bottom of high-rise building in Kuala Lumpur. This is considered as a soft measure in the Malaysian national flood prevention programme (DID 2006). Moreover, as the

maximum inundation depth due to flash flood for a 100-year return period is around 0.2m, which is less than the height of most traditional stilts, the stilts are also an effective way to prevent damage from pluvial flood. The present study shows that such strategy can effectively reduce the flood vulnerability and hence risk for individual buildings. For traditional buildings, which have been altered through time, this feature can be reinstated to restore the traditional character and reduce vulnerability. However, this

solution without proper surface treatment and drainage systems may impact adversely neighbouring buildings.

2.      Improving drainage system and surface condition. Residential buildings which have proper drainage system or vegetation or permeable surrounding ground surfaces or alternatively, set on a higher ground than the road, ensuring a downward slope from the façade to it, were assessed to be in the low

vulnerability class. These conditions are also reflected in the hazard model by varying the percentage of run off in each grid, at a 5 m resolution. Improved drainage systems are recognised as an efficient way to improve the flood resilience of residential buildings without altering their traditional or heritage status. As mentioned above, good drainage is essential for the flood resilience to extend from the single building scale to the urban block to the district.

3. Effectiveness of structural measures. The results obtained highlight that, although the operation of the SMART tunnel can only marginally reduce the spatial extent of the flood and the number of buildings affected, according to the simulation produced in this study, a reduction of about 27% can be observed in the value of the maximum water depth and of about 50% in the cumulative value of losses.

Hence a combination of non structural measures, e.g. use of stilts and proper surface treatment and local

drainage, and structural measures, e.g. SMART, appears to be the most effective strategy to increase flood resilience from building scale to urban scale.

Large major cities in Malaysia, such as Kuala Lumpur, Penang, Petaling Jaya and Shah Alam among others, have been established on floodplains and are increasingly prone to floods and flash-floods as they grow in density and extension (Chan 2011). The use of structural measures is currently under

consideration to address the issue of flooding associated with further urban development. The findings from the present study offer decision-makers an option of increasing building scale resilience, to make structural measures more effective. This is particularly relevant in historical cities such as Penang, where traditional Malay buildings are prevalent. The combination of structural and non-structural measures is also in line with the aspirations of civil society groups that seek urban resilience within

ecological systems (Connolly 2019) and in line with national and international guidelines on flood prevention damage for historic and traditional buildings.

**5. Conclusions**

In this study, a local empirical vulnerability model has been built to evaluate the flood risk to residential buildings in Kampung Baru, Kuala Lumpur. Combining a field survey, Google street view and DEM information, the data of 11 different parameters composing a building level vulnerability model, have been collected and scored to rate the flood vulnerability of a sample of 163 buildings. A new economic loss model is developed to quantify the flood risk in terms of replacement cost, considering both specific vulnerability and a normalised depth-damage ratio function. The flood damage and economic loss were then estimated based on the economic loss model under the flood hazards from 3 different scenarios.

In determining a risk model, a fundamental issue is the level of uncertainty associated to it. In relation to the flood hazard modelling, uncertainty can be identified in the input and the simulation itself. In terms of input, accuracy of water routing is dependent on the DTM accuracy. In the present study a high resolution DTM (0.5m resolution LIDAR) is employed, and checks with aerial imagery and adjustment are made to identify unrealistic flow pathways and amend them. Moreover river locations are defined by analysing the DTM. As a result, the river network may contain false positives, i.e. rivers (and therefore fluvial flood hazard) may be represented in areas where, in reality, there are no streams or watercourses. A second source of input uncertainty is the hydrological input itself, and this is minimised by including in the analysis only gauge data with long and complete records, however it is recognised that gauge data availability in Kuala Lumpur and surrounding areas is poor. Uncertainties in the modelling process arise from two orders of issues: the representation of the flow and the amount of drainage in the model. In relation to the first issue, as each river section is modelled independently, backwater effects at confluences are not represented; furthermore, current individual simulations assumes boundary conditions whereby water can exit the model at the downstream boundary, while in reality if the downstream is also in flood stage, this assumption is not correct. This is an intrinsic limitation of the current fluvial JFlow® model and no mitigation has been implemented for this study. In relation to the overall catchment drainage a fundamental epistemic uncertainty is the location of culverts in Kuala Lumpur, which have not been represented in the model. This is not necessarily a conservative assumption as a blocked culvert may locally exacerbate flooding beyond the level expected in an undefended (no culvert) scenario. Finally, the capacity of natural or artificial drainage systems across the study area is represented at a broad scale and does not fully account for site-specific storm drains or other localised features. A detailed land use dataset was combined with soil information and slope to calculate variable percentage runoff rates on a 30m resolution grid. This resolution is appropriate for the level of detail of the input (land use, soil and slope) information, but means that property-level drainage systems cannot be accounted for.

From the perspective of determining the vulnerability, although increasingly the need for micro level studies is recognised, most published work on flood risk analysis refers to generic building typologies and their incidence on grid-cells containing several buildings, to characterise the exposure. In this respect the vulnerability model proposed here has two advantages: identifies the vulnerability of each

specific assets on the basis of its geometry, material characteristic and level of maintenance, but also in terms of its setting and hydraulic characteristic of its curtilage. This partly compensate the lack of knowledge on drainage feature at the urban scale, from the modelling point of view, but most importantly identifies deficiencies that can be mitigated at the scale of the single property. In developing countries this can become an important tool for communication to stakeholders and

community involvement in mitigation strategies, through the mapping and visualization of the vulnerability indicators. The sample used is relatively small, and although the robustness of the rating process has been verified by cross correlating the scoring results of different surveyors, uncertainties on the single buildings are related to the validity of the Google street map photo and the accuracy with which measurements can be extracted from such pictures. In order to ensure applicability of the

methodology to other locations and to properly calibrate the single parameter's ratings and overall vulnerability classes, larger samples should be studied.

A fundamental source of uncertainties in modelling losses, is the choice of an appropriate damage/depth function, and its conversion in monetary terms. The first is usually mitigated by calibrating any model on damage data for historic floods in the area or region and the second by

calibrating the replacement cost on insurance claim data. In the present study, both historic damage and insurance claim datasets are not readily available in a format that can be used at this scale and in this context. Therefore, rather than using a single arbitrary damage depth function, a large number of functions derived for building types similar to the ones analysed have been used to obtain a mean damage ratio function by regression. This was then validated by comparison with functions derived by

other studies on reach damage datasets. The fact that the damage function is independent of the specific building typology or local exposure model, which are accounted for in the vulnerability model, renders it of generic value and makes it applicable to other situations in Malaysia and worldwide. The economic loss function considers the loss from both the physical damage to each building and its content. The additional cultural value as a touristic attraction was rather crudely accommodated by an arbitrary factor.

There is an extensive, but also so far rather inconclusive debate in literature, as to how to compute and quantify the increase in loss associated with the historic value of a property, both as it pertains to its direct and indirect losses. This is an area that should be tackled in future by looking in detail at the additional repairing costs and the loss in revenue from touristic business. The intangible aspects of course deserve a different approach.


**Acknowledgments**

This study was supported with funding from the Newton Ungku Omar Fund and Innovate UK for the project entitled 'Disaster Resilient Cities: Forecasting Local Level Climate Extremes and Physical Hazards for Kuala Lumpur'. (EP/P015506/1). The authors wish to thank Prof Mark Saunders (UCL

Dept of Space & Climate Physics, Faculty of Maths & Physical Sciences) for providing normalised rainfall profiles and gridded rainfall return levels for the Kuala Lumpur study area, used by JBA to determine the hyetographs for the flood hazard simulations. We would like to thank the two anonymous reviewers for their stimulating and constructive comments.

**Data availability**


Building data were collected from a field survey and Google Street View (https://www.google.com/maps/). Primary data are strictly used within the project "Disaster Resilient Cities: Forecasting Local Level Climate Extremes and Physical Hazards for Kuala Lumpur". The data of the research findings are available from the corresponding author (DDA) on reasonable request.

**Author contributions**

DDA designed the research and analysed the results; KW and YY collected the data, analysed the results and produced the visualisation; HS, AM and VP conducted the flood modelling; JJP discussed and extended the findings. All authors discussed the results and drafted the final manuscript.

**Competing interests**

The authors declare that they have no competing interests.

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
