# Peer review of "Revised manuscript submit to NHESS"

_Natural Hazards and Earth System Sciences, 2020_

## Referee Comment (RC1) · Anonymous Referee #1 · 24 Apr 2020

The authors develop a flood vulnerability method for the assessment of traditional residential buildings in Kuala Lumpur. The study includes a survey of 163 buildings using different building-level vulnerability parameters. This is a very interesting topic that contributes to the recent increase in studies looking at flood vulnerability, damages and mitigation measures at a building level, and it fits very well within the scope of NHESS. In my view, the paper would benefit from an improved explanation of the methods, mainly the parameter selection and valuation, and the findings regarding the vulnerability index (as discussed in more detail below).

Broad comments

- L. 185: what is the proportion and how was it determined? - Section 2.4: I miss a link between (some of) the parameters mentioned in table 2 and the way they impact a

building's flood vulnerability. For example, I understand how footprint influences damages, but how does it link to vulnerability of a building? How does the surface condition link to the vulnerability of a building? The surface condition (permeability / infiltration rate) is commonly perceived as part of the hazard rather than vulnerability (e.g. Liu et al (2014))? It would be good to explain how each of the selected parameters contribute to vulnerability and how you differentiate the extent to which they contribute to vulnerability for each of these parameters. - Section 2.5 (Table 3): many flood building studies differentiate between 1 storey and 2 storeys (e.g Deniz et al. (2016); Englhardt et al. (2019)). Is it realistic to differentiate between an inundation depth up to 3 storeys and 4 storeys or more? Especially because you state that "the maximum inundation depth due to flash flood for a 100-year return period is around 0.2m". - Section 3.1 could be improved by expanding the analysis of the index. E.g. L. 415 states that a normal distribution can be observed from fig 7a. This is not clear and needs to be elaborated on in the text as well as in the figure and its caption. L. 417 states that the total VRi follows a lognormal distribution, while in fig 8 it follows a normal distribution. Next, the caption of 7 mentions "VI", should this then be "VR"? - I think it is very important to emphasize that you are calculating the relative vulnerability. I was initially expecting the vulnerability classes to be categories within the range of 110 (the overall possible minimum) to 1100 (the overall possible maximum). Please elaborate in paragraph at L.320 why this decision was made. - The percentages of the sample column of Table 4 do not add up to 100%. - The abstracts states that: "The paper discusses these in relation to a scenario event of 0.1% Annual Exceedance Probability (AEP), based on hydrological and hydraulic models developed for the Disaster Resilient Cities Project." However, I can't find a mention of this in the body of the manuscript.

Minor things:

- L. 48 "control": not clear what is meant here. - L52 "...political negotiation": these statements look stronger when backed-up with (a) reference(s). - L. 54 maybe include some examples of "Non-structural measures" that provide "faster flood mitigation". -

L. 63 "UNDRR", write the actual name when using the acronym for the first time. - L. 81-83: please add page number(s) of the direct quote (or paraphrase). - L. 89: it would be good to add a reference for the definition of vulnerability. - Figure 2: maybe crop the high rises from 2b so the focus is on the vernacular house. - It may be nice to add a map (or add it to fig 2a) showing the locations of the gauges. - L. 207: along river network of the study area -> the river network - L. 262 ("...by building type"): it would be good to include some references to support this statement. - L. 352 "2415 to 4105 RM (525 to 890 €" -> it would be very useful to add the euro value to each mention of an RM value. - L. 364: typically ranges - L. 408: number of storey -> number of storeys - Fig 7a and L. 415: it is unclear from figure 7a which of the variables represents the roof height. In general, this figure deserves a little bit more explanation and probably best to update the labels with the wording used elsewhere for each of the parameters (same holds for other figures such as fig 10). - L. 424. "and smallest" -> and the smallest - L. 424 "The largest VR is 852.5, and smallest is 477.5." Refer to table 4. - L. 461 "3 different scenarios" -> three different scenarios - L. 474 "total number of building" -> buildings - L. 475. "the total building" -> total number of buildings - Fig 11b caption: SAMRT -> SMART - L. 478: the -> The - L. 483 "without SMART Major losses" -> major - L. 483 "concentrate" -> concentrated - L. 487 "was assessed to have" -> was found to have - Fig 12. The double y-axis is fine, but maybe adjust the colours to improve legibility (e.g. in 12a, the number of flash-flooded buildings and the cumulative graph are around a water depth of 0.1- 0.3 are difficult to decipher). - L. 499. Flood has become a major hazard worldwide. -> better to add a reference for this statement. - L. 501. The word "dearth" is a bit archaic, maybe better to use "lack of" or "limited" - L. 533 "varying the % of run off" -> percentage of run off

---

## Referee Comment (RC2) · Anonymous Referee #2 · 27 Apr 2020

I have now read the paper titled: "Flood Vulnerability Assessment of Urban Traditional Buildings in Kuala Lumpur, Malaysia". The paper focuses on the vulnerability of buildings to flooding in Malaysia by developing a vulnerability index for each building based on a number of parameters and by actually taking a step beyond and calculate also the economic loss under different flood scenarios. The paper presents an interesting approach to vulnerability assessment however it demonstrates also a number of significant weaknesses. In more detail:

Title: The title indicates that the main focus of the paper is the vulnerability assessment of buildings, however, the paper goes beyond that: a hazard map for different scenarios is produced and the possible economic loss under different scenarios is assessed. The title should probably change in order to include all that. Moreover, according to the title

the focus is on traditional buildings, whereas in the abstract the buildings are referred to as urban heritage buildings which indicates something else and elsewhere in the text as residential buildings (page 3, line 101). This should be also considered in rethinking the title of the paper.

Abstract: the abstract is rather long and gives too much detail (e.g. field surveys with Google street view) but also it does not refer to additional aspects that the paper covers such as the economic loss calculation.

Introduction: in the introduction but also elsewhere in the text the authors refer to non-structural measures but they never connect them to the results of their study or their aims. Also in the introduction, they refer to floods but they do not explain what kind of floods they are looking at. Later on in the manuscript, the authors shed light on that matter but it would be better if this would be done earlier on.

SMART: What is the relationship to the authors with the SMART project? Is SMART part of what they are doing or do they just use readymade results from this project? It is not very clear. More clarification is also needed in the description of the SMART defense scenario. What does this include? What kind f defense measures? Where?

Figure 4: The authors estimate the time of peak at all ungauged locations within the study area. Why is this information relevant to the vulnerability assessment of buildings?

Vulnerability index: Why do attributes vary between 3 and 5? Please clarify.

Vulnerability parameters: How do parameters 1 to 6 relate to the expected intensity? I guess 8and this also has to be clarified) that in e.g. parameter 4. With the height of stilts between 0 and 0,5m(?) there is 55 VR. But if the height of the flood is 2m this specific building will be highly vulnerable.

Weighting and classification: the authors do not refer to the weighting of the parameters or the classification of the final VRs. These are two important issues that should be

considered when working with indices. A reference to the following paper which deals with these issues is considered in my opinion necessary:

Papathoma-Köhle M., Schlögl, M., Fuchs, S. 2019. Vulnerability indicators for natural hazards: an innovative selection and weighting approach. Scientific reports.

Flood depth-damage ration function: page 14, lines 375-376: does the window height play a role?

Figure 6: The authors create a vulnerability curve based on the mean values of several damage functions in the literature. Why is it expected that the buildings in Malaysia correspond to an average value of the existing models? The depth damage ration functions used in the paper are from different countries (Japan, Ethiopia, and global generic functions). Clarifications are needed at this point. What are the points in the figure? The building used in the present study. Please clarify.

Figure 7 and 8: The authors present some descriptive statistics of the index. Why is this information relevant? How and by who can it be used?

Table 4: the classification of the vulnerability classes has to be justified.

Interpretation of results: The results are described but not interpreted or used to demonstrate the importance of the approach for specific end-users. For example, (page 17, lines 432-433) "the buildings in the eastern part of the area have higher vulnerability". Why is that (e.g. older part of town?) How can this information be used?

Page 18, lines 454-455: This needs to be discussed more There are two issues here: 1. Why is the number of floors a parameter of flood vulnerability anyway? Is a building with more floors more or less vulnerable to flooding and why? It can offer vertical evacuation to residents but apart from that does the number of floors contribute to the reduction or not of the physical vulnerability? And 2. The high number of floors means high building value which reduces the degree of loss. Some discussion on this kind of drawbacks of the approach is also needed.

Estimation of replacement cost due to different flood scenarios: In my opinion the scenarios should also be reflected in the VR 8see previous comment about vulnerability parameters.

Type of hazard addressed (page 20, line 485): this information comes too late. The authors focus on flash floods and river floods and they combine "the total flood risk". What is the difference between these two processes as far as their impact on the building is concerned? Why do the authors suddenly start talking about risk? Is this what they assess?

Discussion: Some vital information is missing. what were their assumptions and uncertainties? How can this study be improved and further developed in the future? How can the results (e.g. the vulnerability maps) be used by end-users?

Conclusions: the conclusions should be stronger and show what the authors have really achieved with the specific study. Instead there are some repetitions (e.g. lines 564-576) without having a strong message at the end.

Please also note the supplement to this comment:
https://www.nat-hazards-earth-syst-sci-discuss.net/nhess-2020-96/nhess-2020-96-RC2-supplement.pdf

---

## Author Comment (AC1) · 12 Jun 2020

In the following find enclosed comments of reviewer #1 and the authors responses. We thank the reviewer for the thorough reading of the paper and the request for clarifications which have resulted in substantial changes to the manuscript. As a result we believe that the clarity of the manuscript is much improved.

The authors develop a flood vulnerability method for the assessment of traditional residential buildings in Kuala Lumpur. The study includes a survey of 163 buildings using different building-level vulnerability parameters. This is a very interesting topic that contributes to the recent increase in studies looking at flood vulnerability, damages and mitigation measures at a building level, and it fits very well within the scope of

[Figure]

NHESS. In my view, the paper would benefit from an improved explanation of the methods, mainly the parameter selection and valuation, and the findings regarding the vulnerability index (as discussed in more detail below).

Broad comments –L. 185: what is the proportion and how was it determined?

We thank the reviewer for highlighting this statement. At the outset of flood map development, we intended to remove an appropriate portion of gross rainfall to account for the volume of water that storm drains could accommodate. After researching what an appropriate proportion should be, we discovered that there is no clear design standard of drainage in use across the city. We also discovered several media reports stating that urban drainage in the city was ineffective. As a result, we made the decision to use the gross rainfall estimates, without adjusting them, rather than calculating a net rainfall amount to use in the modelling. We have adjusted the manuscript to correct the method description.

- Section 2.4: I miss a link between (some of) the parameters mentioned in table 2 and the way they impact a building's flood vulnerability. For example, I understand how footprint influences damages, but how does it link to vulnerability of a building? How does the surface condition link to the vulnerability of a building? The surface condition (permeability / infiltration rate) is commonly perceived as part of the hazard rather than vulnerability (e.g. Liu et al(2014))? It would be good to explain how each of the selected parameters contribute to vulnerability and how you differentiate the extent to which they contribute to vulnerability for each of these parameters.

Response: In this study, the vulnerability assessment consider the building characteristics and its surrounding environment as a system. The surrounding environment, such as surface condition and drainage system, are closely related to the local permeability and runoff, and impact on the height of water in the case of flooding. These parameters were considered in the vulnerability as current flood models are based on general land use, but they do not consider the immediate surrounding, at the scale of the building,

hence do not include local difference of surfacing and permeability, for instance, which might affect the propensity for vulnerability especially to flash flooding. Text has been edited to include a comprehensive description of the reason for inclusion of the various parameters.

- Section 2.5 (Table 3): many in flood building studies differentiate between 1 storey and 2 storeys (e.g Deniz et al. (2016); Englhardt et al. (2019)). Is it realistic to differentiate between an inundation depth up to 3 storeys and 4 storeys or more? Especially because you state that "the maximum inundation depth due to flash flood for a 100-year return period is around 0.2m".

Response: We agree that the difference between 1 storey and 2 storeys is more significant in terms of the damage to fabric and content. However building with more storeys impose higher pressure on ground and are more susceptible to post-flood subsidence, especially in condition of floodplains and superficial foundations. The flood hazard does not consider only flash flood but also riverine floods. The text has been changed to better explain how the parameter number of storey is considered in this study.

- Section 3.1 could be improved by expanding the analysis of the index. E.g. L. 415 states that a normal distribution can be observed from fig 7a. This is not clear and needs to be elaborated on in the text as well as in the figure and its caption. L. 417 states that the total VRi follows a lognormal distribution, while in fig 8 it follows a normal distribution. Next, the caption of 7 mentions "VI", should this then be "VR"?

Response: The total VI follows a quasi-normal distribution (as shown in figure 8). However the density probability function (or cumulative distribution) of VI follows a LogNormal distribution as shown in figure 7b. The distribution of each parameter has been clarified in Figure 7a and 7b. This portion of the paper has been redrafted to clarify these and other aspects of the description of the index. Throughout the VR and VI have been checked and appropriately referred.

- I think it is very important to emphasize that you are calculating the relative vulnerability. I was initially expecting the vulnerability classes to be categories within the range of 110 (the overall possible minimum) to 1100 (the overall possible maximum). Please elaborate in paragraph at L.320 why this decision was made.

Response: The categories in the manuscript were initially derived from the actual range of the samples. This was made to emphasize the differences within the sample as the range close to the two extreme values are not attained. Nonetheless to show the generic value of the approach, we have recast the results within the full theoretical values, i.e. 110 to 1100 for 11 factors, as suggested by the reviewer, to re-categorise the vulnerability classes. This will make it possible to carry out future comparison with other studies using the same approach. The text and table 4 and subsequent diagrams have been changed accordingly.

- The percentages of the sample column of Table 4 do not add up to 100%.

Response: In accordance with the previous comment, we have changed Table 4, and rechecked the numbers. Thanks.

- The abstracts states that: "The paper discusses these in relation to a scenario event of 0.1% Annual Exceedance Probability (AEP), based on hydrological and hydraulic models developed for the Disaster Resilient Cities Project." However, I can't ïñĄnd a mention of this in the body of the manuscript.

Response: The abstract has been redrafted to shorten it and the relationship to the mentioned project is explained in the introduction L107 to 110 and in the acknowledgements.

Minor things: - L. 48 "control": not clear what is meant here.

- L52 "...political negotiation": these statements look stronger when backed-up with (a) reference(s).

- L. 54 maybe include some examples of "Non-structural measures" that provide "faster ïñĆood mitigation".

Response: This and the above sentences have been redrafted to provide more explanation and references.

L. 63 "UNDRR", write the actual name when using the acronym for the first time. Response: Full name has been added.

- L. 81-83: please add page number(s) of the direct quote (or paraphrase). Response: Page number has been added.

- L. 89: it would be good to add a reference for the definition of vulnerability. Response: definition added

- Figure 2: maybe crop the high rises from 2b so the focus is on the vernacular house. - It may be nice to add a map (or add it to fig 2a) showing the locations of the gauges. Response: Modified.

- L. 207: along river network of the study area -> the river network Response: Modified.

- L. 262 ("...by building type"): it would be good to include some references to support this statement. Response: Reference added

- L. 352 "2415 to 4105 RM (525 to 890 C" -> it would be very useful to add the euro value to each mention of an RM value. Response: Thanks. Good suggestion. added throughout

- L. 364: typically ranges Response: Modified.

- L. 408: number of storey -> number of storeys Response: Modified.

- Fig 7a and L. 415: it is unclear from figure 7a which of the variables represents the roof height. In general, this figure deserves a little bit more explanation and probably best to update the labels with the wording used elsewhere for each of the parameters (same holds for other figures such as fig 10).

Thank you for pointing out the inconsistency. The roof height was not used in this

analysis and it has been removed from figure 5, as it was misleading. Attention has been paid to use the same name and same order for all the parameters in all figures and tables.

- L. 424. "and smallest" -> and the smallest Response: Modified.

- L. 424 "The largest VR is 852.5, and smallest is 477.5." Refer to table 4. Response: Modified.

- L. 461 "3 different scenarios" -> three different scenarios Response: Modified.

- L. 474 "total number of building" -> buildings Response: Modified.

- L. 475. "the total building" -> total number of buildings Response: Modified.

- Fig 11b caption: SAMRT -> SMART Response: Modified.

- L. 478: the -> The Response: Modified.

- L. 483 "without SMART Major losses"->major Response: Modified.

-L.483"concentrate"->concentrated Response: Modified.

-L.487"was assessed to have" ->was found to have Response: Modified.

-Fig12. The doubley-axis is fine, but may be adjust the colours to improve legibility (e.g. in 12a, the number of flash-flooded buildings and the cumulative graph are around a water depth of 0.1- 0.3 are difficult to decipher). Response: Modified.

- L. 499. Flood has become a major hazard worldwide. -> better to add a reference for this statement. Response: Modified.

- L. 501. The word "dearth" is a bit archaic, maybe better to use "lack of" or "limited" Response: Modified.

L. 533 "varying the % of run off" -> percentage of run off Response: Modified.

---

## Author Comment (AC2) · 12 Jun 2020

We thank review 2 for the critical observations relating to the methodology of the paper and the request to expand on the impact and possible use of the study in the community. We believe the reviewer's query were most stimulating and hopefully the answers are equally satisfactory. The manuscript has been amended to reflect these observations and discussion

I have now read the paper titled: "Flood Vulnerability Assessment of Urban Traditional Buildings in Kuala Lumpur, Malaysia". The paper focuses on the vulnerability of buildings to flooding in Malaysia by developing a vulnerability index for each building based on a number of parameters and by actually taking a step beyond and calculate also

the economic loss under different flood scenarios. The paper presents an interest-ing approach to vulnerability assessment however it demonstrates also a number of significant weaknesses. In more detail:

Title: The title indicates that the main focus of the paper is the vulnerability assessment of buildings, however, the paper goes beyond that: a hazard map for different scenarios is produced and the possible economic loss under different scenarios is assessed. The title should probably change in order to include all that. Moreover, according to the title the focus is on traditional buildings, whereas in the abstract the buildings are referred to as urban heritage buildings which indicates something else and elsewhere in the text as residential buildings (page 3, line 101). This should be also considered in rethinking the title of the paper.

Response: We thank the reviewer for the consideration of the relevance of the title to the content of the paper. It is true that we do not only assess vulnerability but we have tried to determine the risks posed to these buildings by 3 different hazards sce-narios. The emphasis of the paper remains on the multi-scale vulnerability, which is novel, rather than on the risk, which is assessed in a more conventional way. For what concerns the buildings, these are indeed traditional, considered as a whole and in this particular setting, they represent an important heritage, within an area which is con-sidered a protected area for minority population settlement, and the specific buildings are residential. So all the above terms apply. We have changed the title and some of the introduction to reflect the reviewer's observation. The new proposed title is: Flood Vulnerability and Risk Assessment of Urban Traditional Buildings in a Heritage District of Kuala Lumpur, Malaysia

Abstract: the abstract is rather long and gives too much detail (e.g. field surveys with Google street view) but also it does not refer to additional aspects that the paper covers such as the economic loss calculation.

The abstract has been updated to reflect the whole content of the paper and shorten
it.

Introduction: in the introduction but also elsewhere in the text the authors refer to non-structural measures but they never connect them to the results of their study or their aims. Also in the introduction, they refer to floods but they do not explain what kind of floods they are looking at. Later on in the manuscript, the authors shed light on that matter but it would be better if this would be done earlier on.

By non-structural measure in this paper we mean adaptive measures at local levels, spatial planning (flood risk adapted land use), building regulation and improvement of building flood resistance (wet-proofing and dry-proofing), flood action plans at a local scale, rather than financial measures such as insurance. Currently there is no sufficient evidence to prove that insurance is an effective measure to mitigate flood risk in Malaysia, to our knowledge. This is why reference to insurance is not made. We have clarified this in the Introduction (lines 51 to 56). We have added a reference to the type of flood analysed in the Introduction. See line 109-112.

SMART: What is the relationship to the authors with the SMART project? Is SMART part of what they are doing or do they just use ready made results from this project? It is not very clear. More clarification is also needed in the description of the SMART defense scenario. What does this include? What kind of defense measures? Where?

SMART (Stormwater Management and Road Tunnel) project is a very well-known major structural flood control intervention, implemented in Kuala Lumpur in the first decade of the 21st century, a first worldwide. The relevant reference is included in the manuscript (Abdullah 2004). It is not within the scope of this manuscript to describe the SMART project in greater details than already included at lines 72 to 74, lines 178 to 181 and Table 1, which clearly explain the operations of the SMART infrastructure and its effect on flooding control in Kampung Baru area.

Figure 4: The authors estimate the time of peak at all ungauged locations within the study area. Why is this information relevant to the vulnerability assessment of buildings?

Response: This has been explained in the text. Lines 237-239.

Vulnerability index: Why do attributes vary between 3 and 5? Please clarify.

Response: Qualitative parameters have 3 attributes, (e.g. Low, medium , high,) while quantitative parameters have 4 to 5 attributes to ensure capture of important quantities which represent thresholds in vulnerability. A sentence has been added to explain this.

Vulnerability parameters: How do parameters 1 to 6 relate to the expected intensity? I guess 8and this also has to be clarified) that in e.g. parameter 4. With the height of stilts between 0 and 0,5m(?) there is 55 VR. But if the height of the flood is 2m this specific building will be highly vulnerable.

Response: in most risk models, hazard and vulnerability are independent variables of the problem. The vulnerability is the propensity of the asset to be damaged given its own characteristics, independently of the magnitude of the hazard. So in this case the vulnerability indicators are independent of the specific intensity of a particular flood with a particular return period in this area, as they are applicable to any other urban context. Therefore in the case of the stilts, the mean value for 0.5 refers to typical values or most probable values of stilts in urban contexts, with direct reference to construction practice. Lower values of the stilt will increase the vulnerability and higher values correspond to lower vulnerability. A building with stilts will still be less vulnerable than a building without. The differential between the flood height and the stilts height is accounted in the damage function as explained in section 3.3. Similarly for other parameters. This explanations have also been extended in the manuscript text.

Weighting and classification: the authors do not refer to the weighting of the parameters or the classification of the final VRs. These are two important issues that should be considered when working with indices. A reference to the following paper which deals with these issues is considered in my opinion necessary: Papathoma-Köhle M.,

Schlögl, M., Fuchs, S. 2019. Vulnerability indicators for natural hazards: an innovative selection and weighting approach. Scientific reports.

It is stated in the manuscript (section 2.5) that all parameters are summed to the VI unweighted as there is not sufficient historical recorded evidence or insurance payment to provide statistical or even anecdotal correlation between specific vulnerability indicator to actual damage or losses so that a classification (ranking) or weighting of any of the parameters would have a statistical significance. For this reason, this strategy is not pursued as already explained also in Stephenson, D'Ayala 2014. Also in fluvial flooding indicators relevance is less polarized than in torrential flooding. A reference to the paper above is included in the text.

Flood depth-damage ration function: page 14, lines 375-376: does the window height play a role? The window sill height has a role, as it can be seen by the steep slope in the region of 0.5 to 1 m. of the damage-flood depth function.Building with a lower window sill have a higher vulnerability than buildings with higher window sill

Figure 6: The authors create a vulnerability curve based on the mean values of several damage functions in the literature. Why is it expected that the buildings in Malaysia correspond to an average value of the existing models? The depth damage ration functions used in the paper are from different countries (Japan, Ethiopia, and global generic functions). Clarifications are needed at this point. What are the points in the figure? The building used in the present study. Please clarify.

Response The curve in Figure 6 is not a vulnerability curve is a damage function. Historically researchers have been using heuristic damage functions derived from historic USA data and recently recast in FEMA MH documents. In recent years other damage functions from other part of the world are emerging, but these depend on available empirical field data. However in most cases such function are obtained as averaged value of insurance claims over grid cells, so the relevance to specific building type or urban conditions is rendered negligible by the averaging. Credible values of flood insurance

claim for Malaysia to derive a robust damage function are not available currently. The reason for using several functions, some global, some local is to eliminate biases of any particular function, by averaging the expected damage ratio for the same flood depth. The high determination coefficient obtained and the relatively modest std for each average point, shows that the process is acceptable, given the lack of more accurate data. A sentence to explain this is added in the text. A statement has been added to clarify that the damage function has been validated against other damage functions derived on the basis of historical damage. Confidence boundary at 95The vulnerability for each individual building is taken care as a multiplier of the damage function in equation 5.

Figure 7 and 8: The authors present some descriptive statistics of the index. Why is this information relevant? How and by who can it be used?

The descriptive statistic is used to validate the empirical model both in terms of the choice of the parameters and the choice of the sample of buildings. For the parameters is seen that they are all differently distributed within the sample, hence they are uncorrelated, which then supports their necessity and sufficiency for inclusion in the vulnerability index model. The cumulative distribution of the VIi shows that the distribution obtained is well represented by a lognormal regression, which again provide confidence in the sample choice to represent the occurrence of different vulnerability level in the district. The descriptive statistics also justify the division of the sample in vulnerability classes (table 4). These are chosen to divide the total vulnerability rating in equal ranges, while identifying threshold values which are critical to the likely response of the building to flood. In terms of who should use this analysis: The vulnerability cumulative function can be used at the level of the single building owner, to determine the level of vulnerability of their property and identify features that can be improved to reduce such vulnerability. At the level of the district and with reference to the map as well as to the classes it can be seen that buildings belonging to the same class are clustered, meaning that there are local intervention at the scale of few compounds, (such as drainage, surfacing, slope) which can be address to reduce such vulnerability.

At the municipal level, if this exercise is repeated for different neighbours then a ranking of them in relation to the mean and dispersion of the VI function can provide support to decision making in terms of nonstructural flood mitigations at neighbor scale. Text has been added to explain this at lines 610-620.

Table 4: the classification of the vulnerability classes has to be justified.

The categories in the manuscript were derived from the actual range of the sample. To make our results more generic, we use the theoretical values, i.e. 110 to 1100 for 11 factors, to re-categorise the vulnerability classes. This makes our methodology and results more comparable with studies conducted in other areas. the manuscript has been modified to reflect this and the classes have been explained.

Interpretation of results: The results are described but not interpreted or used to demonstrate the importance of the approach for specific end-users. For example, (page 17, lines 432-433) "the buildings in the eastern part of the area have higher vulnerability". Why is that (e.g. older part of town?) How can this information be used?

Response: we have included a new discussion and added three examples whch explain the meaning of the results for an individual building owner. At lines 527 to 544.

Page 18, lines 454-455: This needs to be discussed more. There are two issues here: 1. Why is the number of floors a parameter of flood vulnerability anyway? Is a building with more floors more or less vulnerable to flooding and why? It can offer vertical evacuation to residents but apart from that does the number of floors contribute to the reduction or not of the physical vulnerability? And 2. The high number of floors means high building value which reduces the degree of loss. Some discussion on this kind of drawbacks of the approach is also needed.

The text has been changed to better explain how the parameter number of storey is treated in this study. See line 309-313. see also answer to reviewer 1

Estimation of replacement cost due to different flood scenarios: In my opinion the

scenarios should also be reflected in the VR 8see previous comment about vulnerability parameters.

As we already mentioned the vulnerability in this study, as in most other literature on the subject, is independent of the hazard scenario.

Type of hazard addressed (page 20, line 485): this information comes too late. The authors focus on flash floods and river floods and they combine "the total flood risk". What is the difference between these two processes as far as their impact on the building is concerned? Why do the authors suddenly start talking about risk? Is this what they assess?

This comment is related to two earlier comment from this reviewer. We have addressed both the type of flooding and the computation of risk in the title and in the introduction. In the manuscript the risk, or worst case risk scenario for each building was considered on the same map. We agree that this might be confusing and it does not reflect the physical aspect of the phenomenon. For this reason the manuscript has been amended and the risk associated to each of the 3 scenarios produced is evaluated separately and compared to the others.

Discussion: Some vital information is missing. what were their assumptions and uncertainties? How can this study be improved and further developed in the future? How can the results (e.g. the vulnerability maps) be used by end-users? The manuscript has been revised accordingly The possible use by end users is addressed at lines 610 to 658 The assumptions and uncertainties are discussed extensively at lines 660-725

Conclusions: the conclusions should be stronger and show what the authors have really achieved with the specific study. Instead there are some repetitions (e.g. lines 564-576) without having a strong message at the end.

This has been extensively addressed in the conclusions. Line 660-725

2020-96, 2020.

---

## Author Response (AR1)

The authors develop a flood vulnerability method for the assessment of traditional residential buildings in Kuala Lumpur. The study includes a survey of 163 buildings using different building-level vulnerability parameters. This is a very interesting topic that contributes to the recent increase in studies looking at flood vulnerability, damages and mitigation measures at a building level, and it fits very well within the scope of NHESS. In my view, the paper would benefit from an improved explanation of the methods, mainly the parameter selection and valuation, and the findings regarding the vulnerability index (as discussed in more detail below).

We thank the reviewer for the thorough reading of the paper and the request for clarifications which have resulted in substantial changes to the manuscript. As a result we believe that the clarity of the manuscript is much improved.

**Broad comments**

–L. 185: what is the proportion and how was it determined?

Response: We thank the reviewer for highlighting this statement. At the outset of flood map development, we intended to remove an appropriate portion of gross rainfall to account for the volume of water that storm drains could accommodate. After researching what an appropriate proportion should be, we discovered that there is no clear design standard of drainage in use across the city. We also discovered several media reports stating that urban drainage in the city was ineffective. As a result, we made the decision to use the gross rainfall estimates, without adjusting them, rather than calculating a net rainfall amount to use in the modelling. We have adjusted the manuscript to correct the method description.

- Section 2.4: I miss a link between (some of) the parameters mentioned in table 2 and the way they impact a building's flood vulnerability. For example, I understand how footprint influences damages, but how does it link to vulnerability of a building? How does the surface condition link to the vulnerability of a building? The surface condition (permeability / infiltration rate) is commonly perceived as part of the hazard rather than vulnerability (e.g. Liu et al(2014))? It would be good to explain how each of the selected parameters contribute to vulnerability and how you differentiate the extent to which they contribute to vulnerability for each of these parameters.

Response: In this study, the vulnerability assessment consider the building characteristics and its surrounding environment as a system.

The surrounding environment, such as surface condition and drainage system, are closely related to the local permeability and runoff, and impact on the height of water. These parameters were further considered as current flood models are using general land use, but they do not consider the immediate surrounding, at the scale of the building, hence do not include local difference of surfacing and permeability for instance. Text has been edited to include a comprehensive description of the reason for inclusion of the various parameters

- Section 2.5 (Table 3): many flood building studies differentiate between 1 storey and 2 storeys (e.g Deniz et al. (2016); Englhardt et al. (2019)). Is it realistic to differentiate between an inundation depth up to 3 storeys and 4 storeys or more? Especially because you state that "the maximum inundation depth due to flash flood for a 100-year return period is around 0.2m".

Response: We agree that the difference between 1 storey and 2 storeys is more significant in terms of the damage to content. However building with more storeys impose higher pressure on ground and are more susceptible to post-flood subsidence. The flood hazard does not consider only flash flood but also riverine floods. The text has been changed to better explain how the parameter number of storeys is considered in this study.

- Section 3.1 could be improved by expanding the analysis of the index. E.g. L. 415 states that a normal distribution can be observed from fig 7a. This is not clear and needs to be elaborated on in the text as well as in the figure and its caption. L. 417 states that the total VRi follows a lognormal distribution, while in fig 8 it follows a normal distribution. Next, the caption of 7 mentions "VI", should this then be "VR"?

Response: The vulnerability ratings of each parameter follows a normal (or quasi-normal) distribution. However the density probability function (or cumulative distribution) of VI follows a LogNormal distribution as shown in figure 7b. This portion of the paper has been redrafted to clarify these and other aspects of the description of the index.

- I think it is very important to emphasize that you are calculating the relative vulnerability. I was initially expecting the vulnerability classes to be categories within the range of 110 (the overall possible minimum) to 1100 (the overall possible maximum). Please elaborate in paragraph at L.320 why this decision was made.

Response: The categories in the manuscript were derived from the actual range of the samples. This was made to emphasize the differences within the sample as the range close to the two extreme values are not attained. Nonetheless to show the generic value of the approach, we have recast the results within the full theoretical values, i.e. 110 to 1100 for 11 factors, to re-categorise the vulnerability classes. This will make it possible to carry out future comparison with other studies using the same approach. The text and table 4 and subsequent diagrams have been changed accordingly.

- The percentages of the sample column of Table 4 do not add up to 100%.

Response: In accordance with the previous comment, we have changed Table 4, and rechecked the numbers. Thanks.

- The abstracts states that: "The paper discusses these in relation to a scenario event of 0.1% Annual Exceedance Probability (AEP), based on hydrological and hydraulic models developed for the Disaster Resilient Cities Project." However, I can't find a mention of this in the body of the manuscript.

Response: the abstracts has been redrafted to shorten it and the relation to the project is explained in the introduction L107 to 110

**Minor things:**

- L. 48 "control": not clear what is meant here.
- L52 "...political negotiation": these statements look stronger when backed-up with (a) reference(s).
- L. 54 maybe include some examples of "Non-structural measures" that provide "faster flood mitigation".

Response: This and the above sentences have been redrafted to provide more explanation and references.

L. 63 "UNDRR", write the actual name when using the acronym for the first time.

Response: Full name has been added.

- L. 81-83: please add page number(s) of the direct quote (or paraphrase).

Response: Page number has been added.

- L. 89: it would be good to add a reference for the definition of vulnerability.

Response: the definition also refers to Rehman et al 2019

- Figure 2: maybe crop the high rises from 2b so the focus is on the vernacular house. - It may be nice to add a map (or add it to fig 2a) showing the locations of the gauges.

Response: Modified.

- L. 207: along river network of the study area -> the river network

Response: Modified.

- L. 262 ("...by building type"): it would be good to include some references to support this statement.

Response: Reference added

- L. 352 "2415 to 4105 RM (525 to 890 C" -> it would be very useful to add the euro value to each mention of an RM value.

Response: Thanks. Good suggestion.

- L. 364: typically ranges

Response: Modified.

- L. 408: number of storey -> number of storeys

Response: Modified.

- Fig 7a and L. 415: it is unclear from figure 7a which of the variables represents the roof height. In general, this figure deserves a little bit more explanation and probably best to update the labels with the wording used elsewhere for each of the parameters (same holds for other figures such as fig 10).

Response: Thank you for pointing out the inconsistency. The roof height was not used in this analysis and it has been removed from figure 5, as it was misleading. Attention has been paid to use the same name and same order for all the parameters in all figures and tables.

- L. 424. "and smallest" -> and the smallest
Response: Modified.

- L. 424 "The largest VR is 852.5, and smallest is 477.5." Refer to table 4.
Response: Modified.

- L. 461 "3 different scenarios" -> three different scenarios
Response: Modified.

- L. 474 "total number of building" -> buildings
Response: Modified.

- L. 475. "the total building" -> total number of buildings
Response: Modified.

- Fig 11b caption: SAMRT -> SMART
Response: Modified.

- L. 478: the -> The
Response: Modified.

- L. 483 "without SMART Major losses"->major
Response: Modified.

-L.483"concentrate"->concentrated
Response: Modified.

-L.487"was assessed to have" ->was found to have
Response: Modified.

-Fig12. The doubley-axisis fine, but may be adjust the colours to improve legibility (e.g. in 12a, the number of flash-flooded buildings and the cumulative graph are around a water depth of 0.1- 0.3 are difficult to decipher).
Response: Modified.

- L. 499. Flood has become a major hazard worldwide. -> better to add a reference for this statement.
Response: Modified.

- L. 501. The word "dearth" is a bit archaic, maybe better to use "lack of" or "limited"

Response: Modified.

L. 533 "varying the % of run off" -> percentage of run off
Response: Modified. Thanks for all the suggestions.

**Responses to Reviewer 2**

I have now read the paper titled: "Flood Vulnerability Assessment of Urban Traditional Buildings in Kuala Lumpur, Malaysia". The paper focuses on the vulnerability of buildings to flooding in Malaysia by developing a vulnerability index for each building based on a number of parameters and by actually taking a step beyond and calculate also the economic loss under different flood scenarios. The paper presents an interesting approach to vulnerability assessment however it demonstrates also a number of significant weaknesses.

We thank reviewer for the critical observations relating to the methodology of the paper and the request to expand on the impact and possible use of the study in the community. We believe the reviewer's query were most stimulating and hopefully the answers are equally satisfactory. The manuscript has been amended to reflect these observations and discussion.

In more detail:
Title: The title indicates that the main focus of the paper is the vulnerability assessment of buildings, however, the paper goes beyond that: a hazard map for different scenarios is produced and the possible economic loss under different scenarios is assessed. The title should probably change in order to include all that. Moreover, according to the title the focus is on traditional buildings, whereas in the abstract the buildings are referred to as urban heritage buildings which indicates something else and elsewhere in the text as residential buildings (page 3, line 101). This should be also considered in rethinking the title of the paper.
Response: We thank the reviewer for the consideration of the relevance of the title to the content of the paper. It is true that we do not only assess vulnerability but we have tried to determine the risks posed to these buildings by 3 hazards scenarios. The emphasis of the paper remains on the multi-scale vulnerability, which is novel, rather than on the risk, which is assessed in a more conventional way. For what concerns the buildings, these are indeed traditional, considered as a whole and in this particular setting, they represent an important heritage, within an area which is considered a protected area for minority settlement, and the specific buildings are residential. So all the above terms apply. We have changed the title and some of the introduction to reflect the reviewer's observation.

Abstract: the abstract is rather long and gives too much detail (e.g. field surveys with Google street view) but also it does not refer to additional aspects that the paper covers such as the economic loss calculation.
Response: The abstract has been updated to reflect the whole content of the paper and shorten it.

Introduction: in the introduction but also elsewhere in the text the authors refer to nonstructural measures but they never connect them to the results of their study or their aims. Also in the introduction, they refer to floods but they do not explain what kind of floods they are looking at. Later on in the manuscript, the authors shed light on that matter but it would be better if this would be done earlier on.

Response: By non-structural measure in this paper we mean adaptive measures at local levels, spatial planning (flood risk adapted land use), building regulation and improvement of building flood resistance (wet-proofing and dry-proofing), flood action plans at a local scale, rather than financial measures such as insurance. Currently there is no sufficient evidence to prove that insurance is an effective measure to mitigate flood risk in Malaysia, to our knowledge. This is why reference to insurance is not made. We have clarified this in the Introduction (lines 52 to 59). We have added a reference to the type of flood analysed in the Introduction. See line 107-110.

SMART: What is the relationship to the authors with the SMART project? Is SMART part of what they are doing or do they just use readymade results from this project? It is not very clear. More clarification is also needed in the description of the SMART defense scenario. What does this include? What kind of defense measures? Where?

Response: SMART (Stormwater Management and Road Tunnel) project is a very well-known major structural flood control intervention, implemented in Kuala Lumpur in the first decade of the 21$^{st}$ century, a first worldwide. The relevant reference is included in the manuscript (Abdullah 2004). It is not within the scope of this manuscript to describe the SMART project in greater details than already included at lines 75 to 78, lines 190 to 193 and Table 1 in the revised manuscript, which clearly explain the operations of the SMART infrastructure and its effect on flooding controls in Kampung Baru area.

Figure 4: The authors estimate the time of peak at all ungauged locations within the study area. Why is this information relevant to the vulnerability assessment of buildings?

Response: This has been explained in the text. Lines 236-238.

Vulnerability index: Why do attributes vary between 3 and 5? Please clarify.

Response: Qualitative parameters have 3 attributes, (e.g. Low, medium , high,) while quantitative parameters have 4 to 5 attributes to ensure capture of important quantities which represent thresholds in vulnerability. A sentence has been added to explain this. Lines 336-338.

Vulnerability parameters: How do parameters 1 to 6 relate to the expected intensity? I guess 8and this also has to be clarified) that in e.g. parameter 4. With the height of stilts between 0 and 0,5m(?) there is 55 VR. But if the height of the flood is 2m this specific building will be highly vulnerable.

Response: In most risk models, hazard and vulnerability are independent variable of the problem. The vulnerability is the propensity of the asset to be damaged given its own characteristics, independently of the magnitude of the hazard. So in this case the vulnerability indicators are independent of the specific intensity of a particular flood with a particular return period in this area, as they are applicable to any other urban context. Therefore in the case of the stilts, the mean value for 0.5 refers to typical values or most probable values of stilts in urban contexts, with direct reference to construction practice. Lower values of the stilt will increase the vulnerability and higher values correspond to lower vulnerability. A building with stilts will still be less vulnerable than a building without. The differential between the flood height and the stilts height is accounted in the damage function as explained in section 3.3. Similarly for other parameters.

Weighting and classification: the authors do not refer to the weighting of the parametersor the classification of the final VRs. These are two important issues that should be considered when working with indices. A reference to the following paper which deals with these issues is considered in my opinion necessary: Papathoma-Köhle M., Schlögl, M., Fuchs, S. 2019. Vulnerability indicators for natural hazards: an innovative selection and weighting approach. Scientific reports.

Response: It is stated in the manuscript (section 2.5) that all parameters are summed to the VI unweighted as there is not sufficient historical recorded evidence or insurance payment to correlate specific vulnerability indicator to actual damage or losses so that a classification (ranking) or weighting of any of the parameters would have statistical significance. For this reason, this strategy is not pursued as already explained also in Stephenson, D'Ayala 2014. Also in fluvial flooding indicators relevance is less polarized than in torrential flooding. A reference to the paper above is included in the text.

Flood depth-damage ration function: page 14, lines 375-376: does the window height play a role?

Response: The window sill height has a role, as it can be seen by the steep slope in the region of 0.5 to 1 m. of the damage-flood depth function.

Figure 6: The authors create a vulnerability curve based on the mean values of several damage functions in the literature. Why is it expected that the buildings in Malaysia correspond to an average value of the existing models? The depth damage ration functions used in the paper are from different countries (Japan, Ethiopia, and global generic functions). Clarifications are needed at this point. What are the points in the figure? The building used in the present study. Please clarify.

Response: The curve in Figure 6 is not a vulnerability curve is a damage function. Historically researchers have been using heuristic damage functions derived from historic USA data and recently recast in FEMA MH documents. In recent years other damage functions from other part of the world are emerging, but these depend on available empirical field data. However in most cases such function are obtained as averaged value of insurance claims over grid cells, so the relevance to specific building type or urban conditions is rendered negligible by the averaging. Credible values of flood insurance claim for Malaysia to derive a robust damage function are not available currently. The reason for using several functions, some global, some local is to eliminate biases of any particular function, by averaging the expected damage ratio for the same flood depth. The high determination coefficient obtained shows and the relatively modest std for each average point, shows that the process is acceptable, give the lack of more accurate data. A sentence to explain this is added in the text. A statement has been added to clarify that the damage function has been validated against other damage functions derived on the basis of historical damage. The vulnerability is taken care for each individual building as a multiplier of the damage function in equation 5.

Figure 7 and 8: The authors present some descriptive statistics of the index. Why is this information relevant? How and by who can it be used?

Response: The descriptive statistic is used to validate the empirical model both in terms of the choice of the parameters and the choice of the sample of buildings. For the parameters is seen that they are all differently distributed within the sample, hence they are uncorrelated, which then verify their necessity and sufficiency for inclusion in the vulnerability index model. The cumulative distribution of the $VI_i$ shows that the distribution obtained is well represented by a lognormal regression, which again provide confidence in the sample choice to represent the occurrence of different vulnerability level in the district. The descriptive statistics also justify the division of the sample in vulnerability classes (table 4). These are chosen to divide the total vulnerability rating in equal ranges, while identifying threshold values which are critical to the likely response of the building to flood.

In terms of who should use this analysis: The vulnerability cumulative function can be used at the level of the single building owner, to determine the level of vulnerability of their property and identify features that can be improved to reduce such vulnerability. At the level of the district and with reference to the map as well as to the classes it can be seen that buildings belonging to the same class are clustered, meaning that there are local intervention at the scale of few compounds, (such as drainage, surfacing, slope) which can be address to reduce such vulnerability. At the municipal level, if this exercise is repeated for different neighbors then a ranking of them in relation to the mean and dispersion of the VI function can provide support to decision making in terms of nonstructural flood defenses at neighbor scale. Text has been added to explain this at lines 610-620.

Table 4: the classification of the vulnerability classes has to be justified.

Response: The categories in the manuscript were derived from the actual range of the samples. To make our results more generic, we use the theoretical values, i.e. 110 to 1100 for 11 factors, to re-categorise the vulnerability classes. This makes our methodology and results more comparable with studies conducted in other areas.

Interpretation of results: The results are described but not interpreted or used to demonstrate the importance of the approach for specific end-users. For example, (page 17, lines 432-433) "the buildings in the eastern part of the area have higher vulnerability". Why is that (e.g. older part of town?) How can this information be used?

Response: please see new discussion and addition of two examples case which explain the meaning of the results for an individual building owner. At lines 527 to 544.

Page 18, lines 454-455: This needs to be discussed more There are two issues here: 1. Why is the number of floors a parameter of flood vulnerability anyway? Is a building with more floors more or less vulnerable to flooding and why? It can offer vertical evacuation to residents but apart from that does the number of floors contribute to the reduction or not of the physical vulnerability? And 2. The high number of floors means high building value which reduces the degree of loss. Some discussion on this kind of drawbacks of the approach is also needed.

Response: The text has been changed to better explain how the parameter number of storeys is treated in this study. See line 309-313

Estimation of replacement cost due to different flood scenarios: In my opinion the scenarios should also be reflected in the VR 8 see previous comment about vulnerability parameters.

Response: As we already mentioned the vulnerability in this study, as in most other literature on the subject, is independent of the hazard scenario.

Type of hazard addressed (page 20, line 485): this information comes too late. The authors focus on flash floods and river floods and they combine "the total flood risk". What is the difference between these two processes as far as their impact on the building is concerned? Why do the authors suddenly start talking about risk? Is this what they assess?

Response: This comment is related to two earlier comment from this reviewer. We have addressed both the type of flooding and the computation of risk in the title and in the introduction. We do not combine the flood type to compute a total risk. The risk associated to each of the 3 scenarios produced is evaluated separately and compared to the others.

Discussion: Some vital information is missing. what were their assumptions and uncertainties? How can this study be improved and further developed in the future? How can the results (e.g. the vulnerability maps) be used by end-users?

Response: This has been extensively addressed in the conclusions. Line 661-726

Conclusions: the conclusions should be stronger and show what the authors have really achieved with the specific study. Instead there are some repetitions (e.g. lines 564-576) without having a strong message at the end.

Response: This has been extensively addressed in the conclusions. Line 661-726

*Revised manuscript with track change*

[revised manuscript text omitted]

ARCADIS (2019). Construction Cost Handbook MALAYSIA 2019. Accessed December 2019 https://images.arcadis.com/media/7/C/A/%7B7CAC521A-2BD8-4025-AAEB-B3274AE8915F%7DConstruction%20Cost%20Handbook%20Malaysia%202019.pdf

Ashley, R.M.;  Balmforth, D.J.;  Saul, A.J.;  Blansk, J.D. (2005). Flooding in the future–predicting climate change, risks and responses in urban areas. Water Sci. Technol. J. Int. Assoc. Water Pollut. Res.2005,5, 265–273

Balica, S. F., Popescu, I., Beevers, L., & Wright, N. G. (2013). Parametric and physically based modelling techniques for flood risk and vulnerability assessment: a comparison. Environmental modelling & software, 41, 84-92.

BERNAMA (2019). Flash flood inundates PKNS shops in Kampung Baru. Available through http://www.bernama.com/en/news.php?id=1773833

Bhuiyan, T. R., Hasan, M. I., Reza, E. A. C., & Pereira, J. J. (2018). Direct Impact of Flash Floods in Kuala Lumpur City: Secondary Data-Based Analysis. ASM Science Journal, 11(3), 145-157.

CFE-DMHA (Center for Excellence in Disaster Management and Humanitarian Assistance), 2019,

Malaysia Disaster Management Reference Handbook 2019

Chan, N.W. (2011). Addressing Flood Hazards via Environmental Humanities in Malaysia. Malaysian

Journal of Environmental Management 12(2), 11-22

Chen, A. S., Hammond, M. J., Djordjević, S., Butler, D., Khan, D. M., & Veerbeek, W. (2016). From hazard to impact: Flood damage assessment tools for mega cities. Natural Hazards, 82(2), 857-890.

Connolly, C.    (2019). From resilience to multi-species flourishing: (Re)imagining urban environmental governance in Penang, Malaysia. Urban Studies, 1–17, doi: 10.1177/0042098018807573

Custer, R. and Nishijima, K. (2015). Flood vulnerability assessment of residential buildings by explicit damage process modelling. Natural Hazards, 78(1), pp.461-496.

D'Ayala D, Galasso C, Putrino V, Fanciullacci D, Barucco P, Fanciullacci V, Bronzino C, Zerrudo E,

Manolo M, Fradiquela C et al (2016) Assessment of the multi-hazard vulnerability of priority cultural heritage structures in the Philippines. In: Proceedings of the 1st international conference on natural hazards and infrastructure, Chania, Greece, 28–30 June 2016

Department for Irrigation and Drainage (DID), Malaysia (2003) Flood Damage Assessment Of 26 April

2001 Flooding Affecting the Klang Valley and the Generalised Procedures and Guidelines for

Assessment       of       Flood       Damages       (accessed       October       2019)

https://www.water.gov.my/jps/resources/auto%20download%20images/5840fb181a6ea.pdf

Department for Irrigation and Drainage (DID), Malaysia (2006) GUIDELINE ON FLOOD

PREVENTION FOR BASEMENT CAR PARKS

Department for Irrigation and Drainage (DID), Malaysia (2010) MANAGING THE FLOOD

PROBLEM IN MALAYSIA

Dottori, F., Figueiredo, R., Martina, M., Molinari, D. and Scorzini, A. (2016). INSYDE: a synthetic, probabilistic flood damage model based on explicit cost analysis. Natural Hazards and Earth System

Sciences, 16(12), pp.2577-2591.

Dutta D, Herath S, Musiake K (2003) A mathematical model for flood loss estimation. J Hydrol 277:24–

49

Englhardt, J., de Moel, H., Huyck, C. K., de Ruiter, M. C., Aerts, J. C. J. H., and Ward, P. J.:

Enhancement of large-scale flood risk assessments using building-material-based vulnerability curves for an object-based approach in urban and rural areas, Nat. Hazards Earth Syst. Sci., 19, 1703–1722, https://doi.org/10.5194/nhess-19-1703-2019, 2019.

FEMA, 2013. Multi-hazard loss estimation methodology HAZUS-MH - Flood Model Technical

Manual. Department of Homeland Security, Federal Emergency Management Agency, Mitigation

Division, Washington D.C. Department of Homeland Security, Federal Emergency Management

Agency, Mitigation Division, Washington D.C. http://www.fema.gov/media-library-data/20130726-

1820-25045-8814/hzmh2_1_fl_um.pdf

Herbert, D.M., Gardner, D.R., Harbottle, M. et al. (2018) Performance of single skin masonry walls subjected to hydraulic loading. Mater Struct 51: 97. https://doi.org/10.1617/s11527-018-1222-z

Hirabayashi, Y., Mahendran, R., Koirala, S., Konoshima, L., Yamazaki, D., Watanabe, S., ... & Kanae,

S. (2013). Global flood risk under climate change. Nature Climate Change, 3(9), 816.

Howard, A.J., Hancox, E., Hanson J., Jackson R., (2017) Protecting the Historic Environment from

Inland Flooding in the UK: Some Thoughts on Current Approaches to Asset Management in the Light of Planning Policy, Changing Catchment Hydrology and Climate Change, The Historic Environment:

Policy & Practice, 8:2, 125-142, DOI: 10.1080/17567505.2017.1320855

Huizinga, J., Moel, H. de, Szewczyk, W. (2017). Global flood depth-damage functions. Methodology and the database with guidelines. EUR 28552 EN. doi: 10.2760/16510

Inaoka, M., Takeya, K., & Akiyama, S. (2019). JICA's policies, experiences and lessons learned on impacts of urban floods in Asia. International journal of water resources development, 35(2), 343-363.

IPCC, 2013: Climate Change 2013: The Physical Science Basis. Contribution of Working Group I to the Fifth Assessment Report of the Intergovernmental Panel on Climate Change [Stocker, T.F., D. Qin,

G.-K. Plattner, M. Tignor, S.K. Allen, J. Boschung, A. Nauels, Y. Xia, V. Bex and P.M. Midgley (eds.)].

Cambridge University Press, Cambridge, United Kingdom and New York, NY, USA, 1535 pp.

IPCC, 2014 Climate Change 2014: Impacts, Adaptation, and Vulnerability. Part A: Global and Sectoral

Aspects. Contribution of Working Group II to the Fifth Assessment Report of the Intergovernmental

Panel on Climate Change [Field, C.B., V.R. Barros, D.J. Dokken, K.J. Mach, M.D. Mastrandrea, T.E.

Bilir, M. Chatterjee, K.L. Ebi, Y.O. Estrada, R.C. Genova, B. Girma, E.S. Kissel, A.N. Levy, S.

MacCracken, P.R. Mastrandrea, and L.L. White   (eds.)]. Cambridge University Press, Cambridge,

United Kingdom and New York, NY, USA, pp.

Jacobson, C.R. (2011) Identification and quantification of the hydrological impacts of imperviousness in urbancatchments: A review.J. Environ. Manag. 2011,92, 1438–1448 .

Jha, A.K.;   Bloch R.; Lamond, J. (2012). Cities and Flooding:   A Guide to Integrated Urban Flood

Risk Management for the 21st Century; World Bank: Washington, DC, USA, 2012; ISBN 978-0-8213-9477-9.

Kang, S-J, Lee S-J & Lee K-H, (2009) A Study on the Implementation of Non-Structural Measures to Reduce Urban Flood Damage -Focused on the Survey Results of the Experts-, Journal of Asian Architecture and Building Engineering, 8:2, 385-392, DOI: 10.3130/jaabe.8.385

Kelman I, Spence R (2003) A limit analysis of unreinforced masonry failing under flood water pressures. Masonry International 16(2):51–61

Kreibich, H., Van Den Bergh, J. C., Bouwer, L. M., Bubeck, P., Ciavola, P., Green, C., ... & Thieken, A. H. (2014). Costing natural hazards. Nature Climate Change, 4(5), 303.

Kundzewicz, Z.W., et al., 2013. Flood risk and climate change: global and regional perspectives. Hydrological Sciences Journal, 59 (1), 1–28

Kundzewicz, Z. W., Krysanova, V., Dankers, R., Hirabayashi, Y., Kanae, S., Hattermann, F. F., ... & Matczak, P. (2017). Differences in flood hazard projections in Europe–their causes and consequences for decision making. Hydrological Sciences Journal, 62(1), 1-14.

Kundzewicz, Z. W., Su, B., Wang, Y., Wang, G., Wang, G., Huang, J., & Jiang, T. (2019). Flood risk in a range of spatial perspectives–from global to local scales. Natural Hazards and Earth System Sciences, 19(7), 1319-1328.

Lamb, R., Crossley, A. and Waller, S. (2009) "A fast 2D floodplain inundation model." Proceedings of the Institution of Civil Engineers - Water Management, 162, pp. 363-370

Lekuthai A, Vongvisessomjai S (2001) Intangible flood damage quantification. Water Resour Manag 15(5):343–362. doi:10.1023/A:1014489329348

Mallick, R., Tao, M., Daniel, J., Jacobs, J. and Veeraragavan, A. (2015). Development of a methodology and a tool for the assessment of vulnerability of roadways to flood-induced damage. Journal of Flood Risk Management, 10(3), pp.301-313.

Menon Priya (2009). Thousands caught unawares as 2m-high flash floods hit KL. Available through https://www.thestar.com.my/news/nation/2009/03/04/thousands-caught-unawares-as-2mhigh-flash-floods-hit-kl

Merz, B., Kreibich, H., Schwarze, R., and Thieken, A. (2010). Review article "Assessment of economic flood damage", Nat. Hazards Earth Syst. Sci., 10, 1697–1724, doi:10.5194/nhess-10-1697- 2010,.

Milanesi, L., Pilotti, M., Belleri, A., Marini, A., & Fuchs, S. (2018). Vulnerability to flash floods: a simplified structural model for masonry buildings. Water Resources Research, 54(10), 7177-7197.

Min, S. K., Zhang, X., Zwiers, F. W., & Hegerl, G. C. (2011). Human contribution to more-intense precipitation extremes. Nature, 470(7334), 378.

MLIT (2005), Manual for Economic Evaluation of Flood Control Investment (Draft). http://www.mlit.go.jp/river/mizubousaivision/toushin_e/1805_manual_e.pdf

Mohd Hisham, Mohd Anip and Sazali Osman, 2017, FLASH FLOOD FORECASTING AND WARNING IN MALAYSIA, STEERING COMMITTEE MEETING OF THE SOUTHEASTERN ASIA – OCEANIA REGION FLASH FLOOD GUIDANCE JAKARTA, INDONESIA, 10 - 12 JULY 2017

Najibi, N., & Devineni, N. (2018). Recent trends in the frequency and duration of global floods. Earth System Dynamics, 9(2), 757-783.

Nasiri H., Shahmohammadi-Kalalagh, S., 2013, Flood vulnerability index as a knowledge base for flood risk assessment in urban area, Journal of Novel Applied Sciences, 2 (8): 269-272, 2013

Neumann B, Vafeidis AT, Zimmermann J, Nicholls RJ (2015) Future Coastal Population Growth and Exposure to Sea-Level Rise and Coastal Flooding - A Global Assessment. PLoS ONE 10(3): e0118571. https://doi.org/10.1371/journal.pone.0118571

Pall, P., Aina, T., Stone, D. A., Stott, P. A., Nozawa, T., Hilberts, A. G., ... & Allen, M. R. (2011). Anthropogenic greenhouse gas contribution to flood risk in England and Wales in autumn 2000. Nature, 470(7334), 382.

Papathoma-Köhle, M., Schlögl, M., and Fuchs, S.: Vulnerability indicators for natural hazards: an innovative selection and weighting approach, Scientific Reports, 9, Article 15026, https://doi.org/10.1038/s41598-019-50257-2, 2019.

Pistrika, A., Tsakiris, G. & Nalbantis, I. Flood Depth-Damage Functions for Built Environment. Environ. Process. 1, 553–572 (2014). https://doi.org/10.1007/s40710-014-0038-2

Pittore, M., Haas, M., Megalooikonomou, K. (2018): Risk-Oriented, Bottom-Up Modeling of Building Portfolios With Faceted Taxonomies. - Frontiers in Built Environment, 4, 41. http://doi.org/10.3389/fbuil.2018.00041

Prettenthaler , F. , Amrusch, P., Habsburg-Lothringen, C. , (2010) Estimation of an absolute flood damage curve based on an Austrian case study under a dam breach scenario- Nat. Hazards Earth Syst. Sci., 10, 881–894, 2010. https://doi.org/10.5194/nhess-10-881-2010

Rehman, S., Sahana, M., Hong, H., Sajjad, H., & Ahmed, B. B. (2019). A systematic review on approaches and methods used for flood vulnerability assessment: framework for future research.

Natural Hazards, 1-24.

Romali, N.H. et al., (2018) Flood Risk Assessment: A Review Of Flood Damage Estimation Model For

Malaysia, Jurnal Teknologi (Sciences & Engineering) 80:3 (2018) 145–153

Roslan, R., Omar, R. C., Hara, M., Solemon, B., & Baharuddin, I. N. Z. (2019). Flood insurance rate map for non-structural mitigation. In E3S Web of Conferences (Vol. 76, p. 03002). EDP Sciences.

Seo Ryeung Ju, Saari Omar & Young Eun Ko (2012) Modernization of the Vernacular Malay House In

Kampong Bharu, Kuala Lumpur, Journal of Asian Architecture and Building Engineering, 11:1, 95-102,

DOI: 10.3130/jaabe.11.95

Shafiai, S., & Khalid, M. S. (2016). Flood Disaster Management in Malaysia: A Review of Issues of

Flood Disaster Relief during and Post-Disaster. In Int. Soft Sci. Conf (No. 1983, pp. 1-8).

Stephenson, V. and D'Ayala, D. (2014). A new approach to flood vulnerability assessment for historic buildings in England. Natural Hazards and Earth System Sciences, 14(5), pp.1035-1048.

Stone, H., D'Ayala, D., and Wilkinson, S. (2017). The Use of Emerging Technology in Post-disaster

Reconnaissance Missions. London: EEFIT

Stone, H., Putrino, V., and D'Ayala, D. (2018). Earthquake damage data collection using omnidirectional imagery. Front. Built Environ. 4:51 doi: 10.3389/fbuil.2018.00051

Tullos, D. (2018). Opinion: How to achieve better flood-risk governance in the United States.

Proceedings of the National Academy of Sciences, 115(15), 3731-3734.

Wang, G., Wang, D., Trenberth, K. E., Erfanian, A., Yu, M., Bosilovich, M. G. and Parr, D. T. (2017):
The peak structure and future changes of the relationships between extreme precipitation and
temperature. Nature Climate Change, 7 (4), 268–274

Ward, P. J., Jongman, B., Salamon, P., Simpson, A., Bates, P., de Groeve, T., ... Winsemius, H. C. (2015).
Usefulness and limitations of global flood risk models. Nature Climate Change, 5, 712-715.
https://doi.org/10.1038/nclimate2742